# Reducing future air pollution-related premature mortality over Europe by mitigating emissions from the energy sector: assessing an 80% renewable energies scenario

Patricia Tarín-Carrasco[1], Ulas Im[2], Camilla Geels[2], Laura Palacios-Peña[1,3], and
Pedro Jiménez-Guerrero[1,4]

[1]Physics of the Earth, Regional Campus of International Excellence (CEIR) "Campus Mare Nostrum", University of Murcia, Spain.
[2]Aarhus University, Department of Environmental Science, Frederiksborgvej 399, DK-4000, Roskilde, Denmark.
[3]Dept. of Meteorology, Meteored, Almendricos, Spain.
[4]Biomedical Research Institute of Murcia (IMIB-Arrixaca), Spain.

**Correspondence:** Pedro Jiménez-Guerrero (pedro.jimenezguerrero@um.es)

**Abstract.** Overall, European air quality has worsened in the last decades as a consequence of increased anthropogenic emissions, in particular from the sector of power generation. The evidence of the effects of atmospheric pollution (and particularly fine particulate matter, PM2.5) on human health is unquestionable nowadays, producing mainly cardiovascular and respiratory diseases, morbidity and even mortality. These effects can even enhance in the future as a consequence of climate penalties and future changes in the population projected. Because of all these reasons, the main objective of this contribution is the estimation of annual excess premature deaths (PD) associated to PM2.5 on present (1991-2010) and future (2031-2050) European population by using non-linear exposure-response functions. The endpoints included are Lung Cancer (LC), Chronic Obstructive Pulmonary Disease (COPD), Low Respiratory Infections (LRI), Ischemic Heart Disease (IHD), cerebrovascular disease (CEV) and other Non-Communicable Diseases (other NCD). PM2.5 concentrations come from coupled chemistry-climate regional simulations under present and RCP8.5 future scenarios. The cases assessed include the estimation of the present incidence of PD (PRE-P2010), the quantification of the role of a changing climate on PD (FUT-P2010) and the importance of changes in the population projected for the year 2050 on the incidence of excess PD (FUT-P2050). Two additional cases (REN80-P2010 and REN80-P2050) evaluate the impact on premature mortality rates of a mitigation scenario in which the 80% of European energy production comes from renewables sources. The results indicate that PM2.5 accounts for nearly 895,000 [95% confidence interval (95% CI) 725,000-1,056,000] annual excess PD over Europe, with IHD being the largest contributor to premature mortality associated to fine particles in both present and future scenarios. The case isolating the effects of climate penalty (FUT-P2010) estimates a variation +0.2% on mortality rates over the whole domain. However, under this scenario the incidence of PD over central Europe will benefit from a decrease of PM2.5 (-2.2 PD/100,000 h.) while in Eastern (+1.3 PD/100,000 h.) and Western (+0.4 PD/100,000 h.) Europe PD will increase due to increased PM2.5 levels. The changes in the projected population (FUT-P2050) will lead to a large increase of annual excess PD (1,540,000, 95% CI 1,247,000-1,818,000), +71.96% with respect to PRE-P2010 and +71.67% to FUT-P2010) due to the aging of the European population. Last, the mitigation scenario (REN80-P2050) demonstrates that the effects of a mitigation policy increasing the ratio of renewable sources

in the energy mix energy could lead to a decrease of over 60,000 (95% CI 48,500-70,900) annual PD for the year 2050 (a decrease of -4% in comparison with the no-mitigation scenario, FUT-P2050). In spite of the uncertainties inherent to future estimations, this contribution reveals the need of the governments and public entities to take action and bet for air pollution mitigation policies.

## 1 Introduction

Air pollution is nowadays a leading cause of global disease burden, especially in low- and middle-income countries (Balakrishnan et al., 2019), and is expected to greatly increase under future climate scenarios (e.g. Fang et al. (2013a); Tarín-Carrasco et al. (2019); Park et al. (2020), among others). Fine particulate matter (PM2.5) is a common air pollutant with important effects on human health. Exposure to this pollutant leads to cardiovascular or respiratory diseases, together with an increase in premature mortality (e.g. (Brook et al., 2010; Evans et al., 2013; Hamra et al., 2014; Ford and Heald, 2016; Im et al., 2018; Tarín-Carrasco et al., 2019), among others). Short- or long-term exposure to PM2.5 can have different impacts on human health. The much larger effects of long-term exposure may suggest that the effects on human health are not only due to increased pollution, but also to the progression of underlying diseases (World Health Organization, 2013).

In addition, over 90% of the population who lives in cities is exposed to fine particles in concentrations exceeding the air quality guidelines established by World Health Organization (WHO) (Prüss-Üstün et al., 2016). Lelieveld et al. (2013) estimate that 69% of the global population is exposed to an annual mean anthropogenic PM2.5 concentration $>10$ $\mu$g m$^{-3}$ (WHO air quality guideline); 33% to concentrations over 25 $\mu$g m$^{-3}$ (limit value of EU Directive 2008/50/CE); and 20% to concentrations $>35$ $\mu$g m$^{-3}$, the WHO Level 1 Interim Target (World Health Organization, 2013). Focussing on Europe for present scenarios, Lelieveld et al. (2013) calculate global respiratory mortality incidence associated to air pollution as 773,000 per year. The same study provides a burden of 186,000 premature deaths per year associated to lung cancer and a mortality incidence around 2,000,000 caused cardiovascular diseases. For Europe, Andersson et al. (2009) estimate the number of excess premature deaths (PD) as 301,000 per year caused by PM2.5.

Nowadays, 70% of the globally mortality attributable to air pollution is associated to PM2.5 (Silva et al., 2016a), with some hotspots in East Asia, India and Europe. However, mortality attributable to air pollution has changed over the last 25 years (Fang et al., 2013a; Silva et al., 2013; Cohen et al., 2017). Silva et al. (2013) attribute these increases in mortality to direct changes in anthropogenic emissions and estimates that 2.1 million of Chronic Obstructive Pulmonary Disease (COPD) and Lung Cancer (LC) premature deaths are related to PM2.5. In addition, these numbers are expected to increase under future climate scenarios as a consequence of the effect of the *climate penalty* on air quality (Silva et al., 2016b; Hong et al., 2020; Park et al., 2020). Climate change will modify air quality by altering physico-chemical processes and parameters such as temperature (and thus the oxidative capacity of the atmosphere), wet deposition or dynamical changes (Jacob and Winner, 2009; Jiménez-Guerrero et al., 2013a).

Despite fine particulate matter can travel long distances, provoking the increase of mortality on a global scale, Anenberg et al. (2014) estimate that 93%-97% of PD associated to air pollution occur within the source region. Therefore, the contribution of

anthropogenic emissions to air pollution is remarkable. As estimated by Fang et al. (2013a), the 95% of mortality from PM2.5 is driven by local emissions of short-lived air pollutants and their precursors. The main source of emissions responsible for these numbers differs among regions. For instance, in Europe agriculture is the sector with the highest contribution to PM2.5 emissions (Lelieveld et al., 2015; Crippa et al., 2019). However, the sources responsible for the largest impact on PD linked to outdoor air pollution are not related to agriculture, but to land traffic and energy use (Lelieveld et al., 2015; Silva et al., 2016a).

Therefore, the implementation of mitigation controls and environmental policies that can help offsetting the effect of climate penalty become essential for reducing premature mortality over Europe (McConnell et al. (2006); Anenberg et al. (2014); Fang et al. (2013); Crippa et al. (2019), among others). Changes in future anthropogenic emissions will depend on different variables; such as socioeconomics, technology and developments, energy demand, demographic trends and land use change, as well as climate policies (Kirtman et al., 2013). In this sense, Silva et al. (2016a) suggested that specific actions targeting in residential and commercial sectors can control the emissions on PM2.5 and would benefit human health. Other works, as those of Anenberg et al. (2014) or Liang et al. (2018) showed that reducing anthropogenic emissions by 20% can substantially decrease the incidence of excess mortality. Under a business as usual scenario (no emission control), the contribution of outdoor pollution to PD could increase by 100% by the mid-century, doubling in 2050 (Lelieveld et al., 2015). In this line, Lelieveld et al. (2019) showed that replacing fossil fuels by renewable energy sources could improve the numbers related to the loss of life expectancy from air pollution. However, previous studies have shown that e.g. an aging population in the future, have the potential to counteract the effect of these emission reductions (Geels et al., 2015).

Hence, the objective of this study is to estimate the present (1991-2010) incidence of excess PD (PD per year) associated to fine particulate matter and their changes under several future scenarios for the years 2031-2050 that include climate penalty, projected population by 2050 and a mitigation scenario where the 80% of the European energy production comes from renewable sources. A number of different endpoints or causes of premature mortality as Lung Cancer (LC), Chronic Obstructive Pulmonary Disease (COPD), Low Respiratory Infections (LRI), Ischemic Heart Disease (IHD), cerebrovascular disease (CEV) and other Non-Communicable Diseases (other NCD) is included in this contribution.

## 2 Methodology and data

### 2.1 Premature mortality estimation by exposure-response functions

Future PD caused by several specific endpoints related to PM2.5 have been estimated using non-linear exposure-response functions, an analogous methodology as that previously implemented in Tarín-Carrasco et al. (2021). The health impact function in each grid cell has been applied (Equation 1) to estimate premature mortality:

$$\Delta M = y_0 \times \left[ \frac{RR - 1}{RR} \right] \times Population \tag{1}$$

This equation is based on epidemiological relationships between air pollution concentration and mortality in each grid cell, where $\Delta M$ is premature mortality due to a specific disease, $y_0$ is the baseline mortality rate, $RR$ is the risk ratio and

*Population* refers to the exposed population (in this contribution, adults are considered). $y_0$ varies according to the mortality cause, age and European region (Figure SM1 in the Supplementary Material) and is estimated by the WHO for each sex and every year. Sex mixing values used in the present study account for both male and female dwellers during the year 2017 (the last available). Premature mortality and RRs has been estimated for each pathology and different group ages included in this contribution: 25-29, 30-34, 35-39, 40-44, 45-49, 50-54, 55-59, 60-64, 65-69, 70-74, 75-79, +80 and all ages.

Risk ratios were determined following the GEMM methodology developed by Burnett et al. (2018) (Equation 2):

$$RR = exp\left[\theta \frac{log[\frac{z}{\alpha}+1]}{(1+exp(-\frac{z-\mu}{\nu}))}\right], where\ z = max(0, PM2.5 - 2.4\mu gm^{-3}) \tag{2}$$

where $\theta$, $\alpha$, $\mu$ and $\nu$ are variables obtained from Burnett et al. (2018) for each pathology and $z$ refers to the PM2.5 mean annual concentration. The pathologies included in this work are Lung Cancer (LC), Chronic Obstructive Pulmonary Disease (COPD), Lower Respiratory Infection (LRI), Cerebrovascular Disease (CEV), Ischemic Heart Disease (IHD) and non-accidental diseases (NCD+LRI). "Other NCD" is calculated as the subtraction of NCD+LRI and the rest of the categories. Uncertainty ranges are expressed as the 95% confidence intervals (95% CIs), adopted from Burnett et al. (2018).

## 2.2  Population data

Population data for Europe has been taken from the NASA SocioEconomic Data and Applications Center (2019) gridded dataset. These data provide the population density by age and gender for the year 2010 consistent with national censuses and population registers with a resolution of 5 km$^2$. Population data were interpolated to the working grid to make it consistent with the gridded air pollution data (Figure SM2,top in the Supplementary Material).

With respect to the future population, a projection for the year 2050 has been estimated by using information from the Population Prospects from United Nations Organization (UN) Department of Economic and Social Affairs Population Dynamics (United Nations, 2020). This includes both a development of the total national numbers but also the age distribution. The relative variation of the population from this dataset between 2010 and 2050 for each European country and age range was calculated in order to obtain the ratio of population for the future scenario (2050) in this study (Figure SM2,bottom in the Supplementary Material). The population pyramid both for the year 2010 and 2050 is presented in Figure 1. This Figure indicates a very slight projected decrease (-0.2%) of the European population (807.5 M vs. 806.2 M dwellers both for present and future population, respectively), especially over Eastern Europe (-4.0%). Conversely, population in Western and Central Europe increases in the UN 2050 projections (+2.7% and +5.7%, respectively). In addition, the projected data includes a higher population density over many urban areas, and a clear aging of the European citizens. As an example, population over 80 years (80+) barely represents 4% of the total European population nowadays, while it is expected to increase to >9% in the projected UN 2050 estimations.

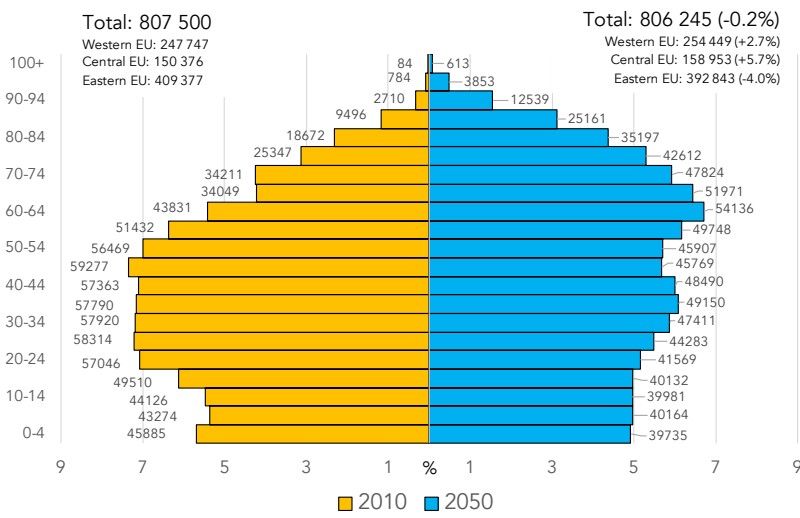

**Figure 1.** Population pyramid by age range for the year 2010 (orange, left) and 2050 projected by the UN (blue, right). Population data are shown in thousands. x-axis represents the percentage of the contribution of each age range to the total population.

## 2.3 Air quality data and scenarios

The availability of observed air pollution data for conducting studies on the impacts of air pollution on human health is scarce. The network of stations for measuring air pollutants is generally insufficient for health purposes due to their spatial misalignment and low coverage (Vedal et al., 2017). This limitation leads to the use of modelling outputs for providing information about air pollution, especially if future air quality projections are needed (Tarín-Carrasco et al., 2019).

Here, air quality model data (PM2.5 dry aerosol mass) from the WRF-Chem model (Grell et al., 2005) under the REPAIR initiative (Jerez et al., 2020; Palacios-Peña et al., 2020b; Pravia-Sarabia et al., 2020; Jerez et al., 2021; López-Romero et al., 2021) is used as input to Equation 2 in order to estimate the annual PD associated to different endpoints in both current and future climate change scenarios. The parameterizations implemented in the WRF-Chem model are summarized in Table SM1 of the Supplementary Material. The domain covers Europe with a horizontal resolution of 0.11° under the Euro-CORDEX requirements (Jacob et al., 2020). For future scenarios, climate forcing is derived from the RCP8.5 scenario, since RCP8.5 represents an upper limit to climate impacts (Moss et al., 2010). The reference periods span 1991-2010 for the present and 2031-2050 for the future projections (PM2.5 concentrations are averaged together for each 20-year period). The robustness of this simulation for representing PM2.5 is evaluated in the Supplementary Material (Tables SM2 and SM3), where the model has been compared with data from 108 stations belonging to the AirBase database of the European Environment Agency. The results are summarized in Figure 2, and the numerical results for each station can be found in Table SM3. Briefly, the low

errors found (for example, average mean bias under 2 $\mu$g m$^{-3}$ and mean fractional bias < 9%) guarantee the phase accordance (timing) between the simulated and observational series, their similar amplitude and, also, the quantitative accuracy of the simulated climatologies, hence making us confident of the suitability of the modeling system for the purpose of this study.

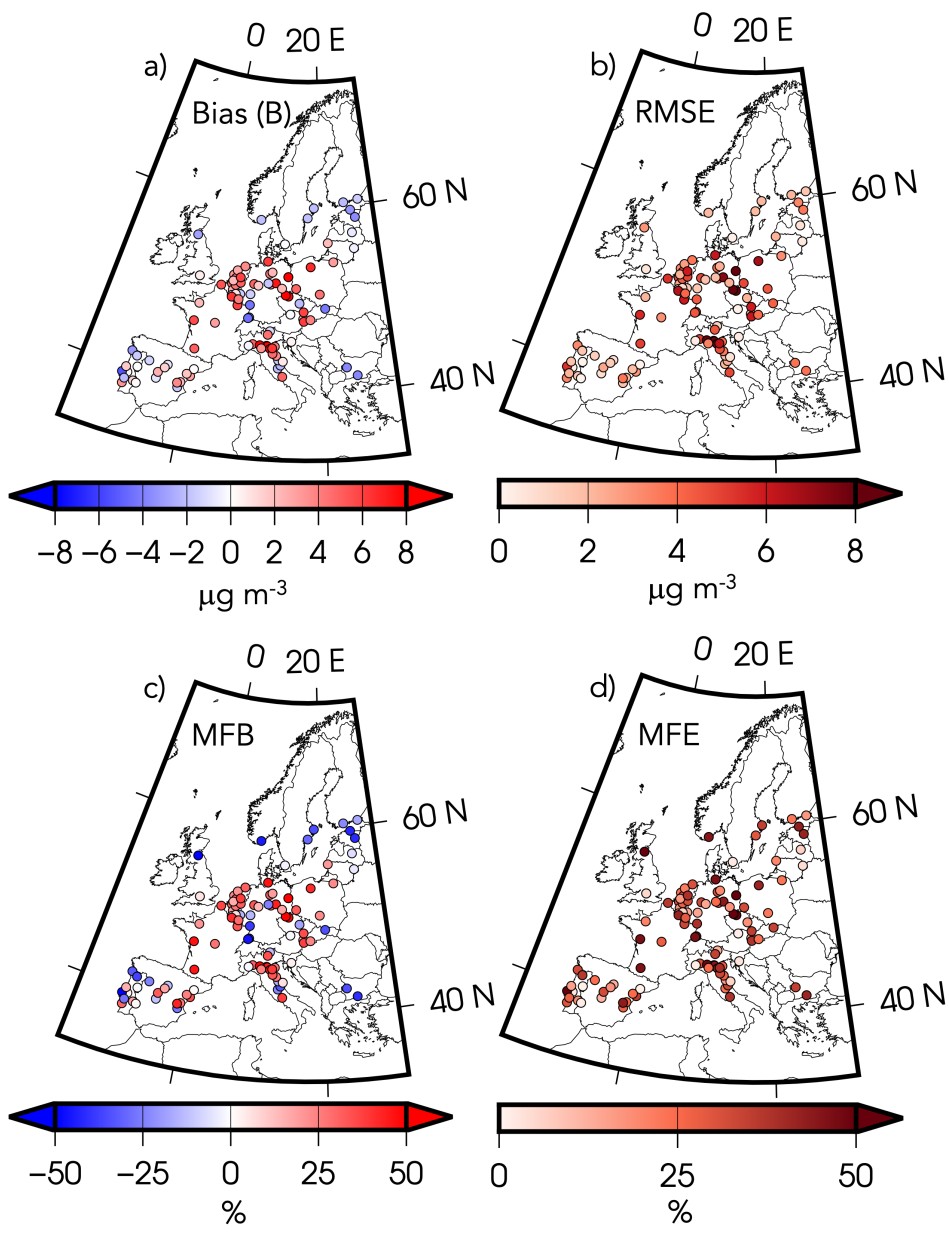

**Figure 2.** Results of the model validation for PM2.5 simulations: (a) mean bias (B, $\mu$g m$^{-3}$); (b) root mean square error (RMSE, $\mu$g m$^{-3}$); (c) mean fractional bias (MFB, %); (d) mean fractional error (MFE, %).

In our case, simulations do not assimilate observational data. Ground-based observations and satellite products are often used to improve modeling results for present-day simulations concerning particulate matter (e.g. Lee et al. (2015); van Donkelaar et al. (2016); Chen et al. (2020); Jiménez-Guerrero and Ratola (2021); McDuffie et al. (2021). However, these bias-correction techniques, widely used in climate impact modeling (Maraun, 2016), are limited when future scenarios are included in the simulations, since no observations can constrain future modeling results. Instead, we have decided to use the so-called "delta method" (Räisänen, 2007) to present the results and the future changes in air pollution, as recommended in Fernández et al. (2019). In the simple terms applied in this contribution, we assume that the results of the evaluation presented in the Supplementary Material point to accurate results (small biases) for present-day PM2.5 simulations. In addition, the delta-method assumes that model errors for future time slice (2031-2050 in this contribution) will cancel out when compared with the present climate simulation (1991-2010, taken as a present reference time slice). This is related to bias correction methods. In particular, delta changes are insensitive to local shift bias correction methods. It is true that more complex bias-correction techniques could have been applied (e.g. quantile mapping), but for those methods, bias corrected and delta change projections differ (Ho et al., 2012; Räisänen and Räty, 2013; Fernández et al., 2019), leading to a new source of uncertainty. Therefore, this contribution uses the delta method (assuming the cancelation of present and future biases), as also implemented in other works related to air pollution impacts on health issues (e.g. Silva et al. (2017), or the contributions of Tarín-Carrasco et al. (2019); Tarín-Carrasco et al. (2021); Guzmán et al. (2022) that rely on these very simulations; among many others).

Further details about the methodology of the model simulations are included below. The GOCART aerosol module (Ginoux et al., 2001; Chin et al., 2002), the aerosol scheme used in this work, includes a bulk approach for black carbon (BC), organic carbon (OC), and sulfate, as well as a sectional scheme for mineral dust and sea salt using Kok (2011) brittle fragmentation theory, a simple and cheap computational approach (Palacios-Peña et al., 2020a). In this work, this scheme has been coupled with the RACM-KPP (KPP: kinetics preprocessor; Stockwell et al. (1997); Geiger et al. (2003). ISORROPIA (Nenes et al., 1998) was used for thermodynamic partitioning of aerosols.

In order to isolate the possible effects of climate change on pathologies only due to changes in atmospheric pollutants, constant anthropogenic emissions for all present and future simulations are assumed. Anthropogenic emissions come from the ACCMIP database (Lamarque et al., 2010) for the year 2000 by country and sector with a spatial resolution of $0.1°$. This allows possible impacts to be anticipated if mitigation strategies for regulatory pollutants are not carried out and characterizes the climatic penalty on air quality levels. ACCMIP compiled a global emission dataset with annual official or scientific inventories at the national or regional scale for $CH_4$, NMVOC, CO, $SO_2$, $NO_x$, $NH_3$, PM10, PM2.5, black carbon and organic carbon. Climate-dependent natural emission sources include desert dust, sea salt aerosols and biogenic volatile organic compounds (VOCs). The emissions were pre-processed according to Freitas et al. (2011).

As stated in Ukhov et al. (2021), the estimation of the PM2.5 is carried out by the subroutine *sum_pm_gocart* in *module_gocart_aerosols.F*. This estimation considers dust and sea salt concentration in their bins 1 (ranges 0.1-1.0 and 0.1-0.5 $\mu$m, respectively) and 2 (1.0-1.8 and 0.5-1.5 $\mu$m, respectively), black and organic carbon and sulphate. GOCART does not include the treatment of secondary organic aerosols (SOA). The authors are aware of limitation; however, the WRF-Chem

version forces to use the GOCART scheme if desert dust and sea salt aerosols are to be included (Palacios-Peña et al., 2020a).
Nitrate aerosols are also not explicitly included in the simulations conducted here.

Last, it should be mentioned that the GOCART aerosol scheme in the WRF-Chem simulations presented here does not allow a full coupling of aerosol-cloud interactions (Palacios-Peña et al., 2020b). For instance, convective wet scavenging and cloud chemistry are not available. However, here the Morrison microphysics (Morrison et al., 2009) acts as a double moment scheme. Hence, the configuration of the model here allows a double-moment microphysics with greater flexibility when representing size distributions and hence microphysical process rates (Palacios-Peña et al., 2020a). When the double moment scheme is activated (as here), a prognostic droplet number concentration using gamma functions and mixing ratios of cloud ice, rain, snow, graupel and hail, cloud droplets, and water vapor is estimated (Morrison et al., 2009). Finally, the interaction of cloud and solar radiation with the Morrison microphysics scheme is implemented in WRF-Chem. Therefore, the droplet number will affect both the droplet mean radius and the cloud optical depth calculated by the model, affecting cloud and precipitation in the model.

### 2.3.1 Emissions scenarios

As aforementioned, anthropogenic emissions come from the ACCMIP initiative (Lamarque et al., 2010) for the year 2000. Taking those emissions as basis, a scenario where emissions from the energy sector in the future have been mitigated is defined, based on the *European 2050 Roadmap* (European Climate Foundation, 2010) of the European Climate Foundation (ECF). The ECF sets three possible scenarios in their strategy for the year 2050, which differ on the percentage of renewable energy production (40%, 60% and 80%) aimed in 2050. Despite complicated, the ECF indicates that the 80% scenario considered here is achievable if the power sector is assumed to implement essentially carbon-free technologies, and hence that emission scenario (from now on, denoted as REN80) has been implemented as a mitigation scenario in this contribution.

Present-day energy production has been estimated from data of the European Environment Agency (2020) with respect to gross electricity production by fuel (Table 1). For the REN80 scenario, energy production by different energy sectors like coal, gas and nuclear are taken into account as the 20% of non-renewable energies. The remaining 80% is expected to be produced from sources such as wind power (representing almost one third of the energy production in 2050 with a percentage of 30%); solar power (23%); biomass and hydropower (12% each); and geothermal, which presents the smallest contribution in 2050 (2%). Figure 3 shows the energy mix by sector for the year 2050 in REN80. For this scenario, the present and future power production in the target domain is presented in Table 1.

In order to estimate emissions from energy production, the emission factors were obtained from the EMEP/EEA air pollutant emission Inventory Guidebook – 2019 (European Environment Agency, 2019). The emission factors from coal (brown, coking, steam, sub-bituminous and hard), natural gas, gaseous fuel, residual oil and gas oil energy production were selected based on the activity data from Tier 2 method in section 1.A.1.a "Public electricity and heat production" within the chapter 1.A.1. "SNAP 01 Combustion in energy and transformation industries" of "Part B: sectoral guidance chapters". Figure 4 shows the emission factors for coal, oil and gas. The highest emission factors for CO, NOx and SOx are related to coal (1174 g/GJ; 3,170 g/GJ and

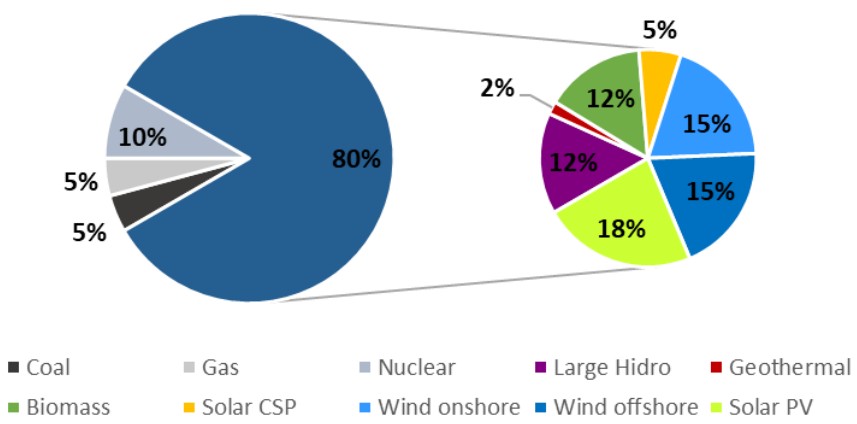

**Figure 3.** Energy mix in the REN80 scenario for the year 2050.

**Table 1.** Annual energy production (in billion of GJ) over Europe from different sources and contribution to the energy mix in present and REN80 scenario.

|  | Present | (1991-2010) | Future | (2031-2050) |
| --- | --- | --- | --- | --- |
|  | $GJ \times 10^9$ | % | $GJ \times 10^9$ | % |
| Renewables | 5.15 | 29.23% | 14.10 | 80.00% |
| Nuclear | 4.54 | 25.80% | 1.76 | 10.00% |
| Oil | 0.32 | 1.83% | – | – |
| Coal | 3.74 | 21.25% | 0.88 | 5.00% |
| Gas | 3.48 | 19.73% | 0.88 | 5.00% |
| Other fuels | 0.38 | 2.16% | – | – |

11,640 g/GJ, respectively), while oil is the most important contributor (per GJ of energy produced) in the case of particulate matter PM10 and PM2.5 (923 g/GJ and 798 g/GJ, in that order) and non-methane volatile organic compounds (436 g/GJ).

Finally, present and future emissions from the energy sector were estimated following the European Environment Agency
(2019) methodology (Equation 3) from the energy produced by each type of fuel and the corresponding emission factor:

$$Emissions = Activity\ Factor(GJ) \times Emission\ Factor(g/GJ) \tag{3}$$

The annual mass of pollutants emitted by the different fuels included here (coal and lignite, oil and gas) is shown in Table 2. The estimations indicate that 6 Mtons of emissions of pollutants are annually saved in the REN80 emission scenario compared to the baseline present emissions. Focusing on PM2.5 (the main aim of this contribution), annual emissions decrease from 0.22
Mtons to 0.05 Mtons (reduction of around -79% in the REN80 scenario). Primary anthropogenic PM2.5 (PPM2.5) emissions

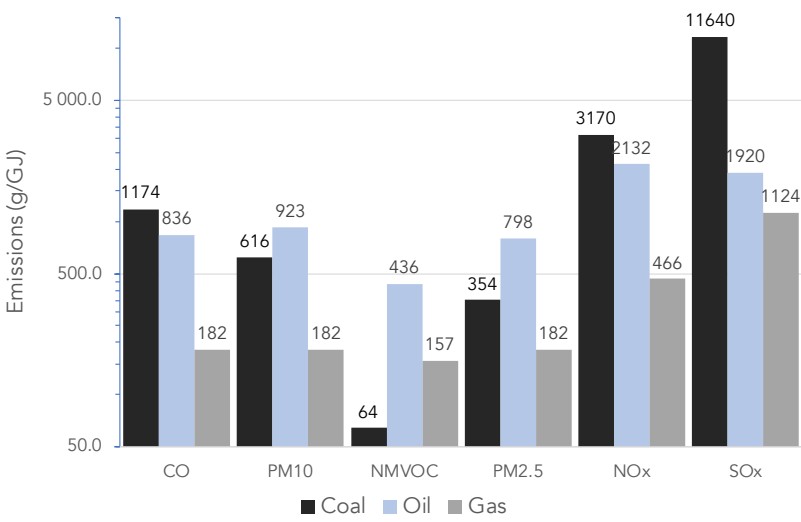

**Figure 4.** Emission factors (g/GJ) by different type of fuel.

is speciated as follows: 7.5% of PPM2.5 corresponds to black carbon; 55.6% to organic carbon and 36.5% to inert PPM2.5. Further information about PM emissions in Europe can be found in Im et al. (2015).

**Table 2.** Annual 2010 ACCMIP emissions of regulatory air pollutants produced by the energy sector used in the simulations (in Mtons) for the SNAP sector representing energy production in the present scenario and projected future emissions under a mitigation scenario taking ACCMIP present emissions as basis. Last column indicates the total emissions in the domain from all sectors.

|  | Present (1991-2010) | Future (2031-2050) | ΔEmissions | All sectors (1991-2010) |
|---|---|---|---|---|
| CO | 0.53 | 0.12 | -77.4% | 3.58 |
| NMVOC | 0.09 | 0.02 | -78.5% | 8.32 |
| NOx | 1.42 | 0.32 | -77.4% | 9.91 |
| PM10 | 0.33 | 0.07 | -78.4% | 2.34 |
| PM2.5 | 0.22 | 0.05 | -78.7% | 1.52 |
| SOx | 4.81 | 1.13 | -76.5% | 8.20 |
| TOTAL | 7.73 | 1.71 | -77.9% | 33.87 |

### 2.3.2 Cases for the estimation of present and future premature deaths over Europe

Table 3 compiles the different cases that have been included in this contribution for the estimation of PD associated to PM2.5 in Europe. First, the PRE-P2010 case uses present day annual mean concentrations of PM2.5 (1991-2010) and population corresponding to the present period (2010) to estimate PD for baseline conditions. In order to isolate the climate penalty, the FUT-P2010 case uses future concentrations of PM2.5 under the RCP8.5 scenario, keeping population at 2010 levels. Introducing the UN 2050 population changes in the case FUT-P2050 allows estimating the variation of PD caused by modifications in the population pyramid over Europe. Last, REN80-P2010 and REN80-P2050 use the modelled PM2.5 concentrations of pollutants for the future RCP8.5 scenario using REN80 emissions and with population corresponding to the year 2010 and changes for 2050, respectively.

**Table 3.** Summary of cases considered for the estimation of premature deaths over Europe.

| Acronym | Period | Forcing | Population | Emissions |
| --- | --- | --- | --- | --- |
| PRE-P2010 | 1991-2010 | MPI-M historical[*] | 2010 | ACCMIP |
| FUT-P2010 | 2031-2050 | MPI-M CMIP5 rcp85[**] | 2010 | ACCMIP |
| FUT-P2050 | 2031-2050 | MPI-M CMIP5 rcp85[**] | 2050 | ACCMIP |
| REN80-P2010 | 2031-2050 | MPI-M CMIP5 rcp85[**] | 2010 | ACCMIP modified according REN80 |
| REN80-P2050 | 2031-2050 | MPI-M CMIP5 rcp85[**] | 2050 | ACCMIP modified according REN80 |

[*]Giorgetta et al. (2012a); [**]Giorgetta et al. (2012b)

## 3 Results and discussion

The results presented in this section try to disentangle the impacts of PM2.5 levels over Europe on present and future premature mortality over Europe. For that purpose, a brief description of the changes in PM2.5 concentration as a consequence of the climate penalty are presented. Once these changes are established, the estimation of present PD (PRE-P2010 case) over three different areas of Europe (Western EU, Central EU and Eastern EU) are discussed. Next, the effect of climate penalty on PD is assessed for a future scenario (RCP8.5, 2031-2050) keeping the population constant at 2010 levels (FUT-P2010). The results for the future case considering also the change on the population for 2050 (FUT-P2050) have been studied and compared with FUT-P2010. Finally, the effects of a future mitigation scenario are quantified. That scenario includes the use of 80% of renewables sources to produce energy (REN80-P2010 and REN80-2050) and allows to isolate the effect of a future mitigation strategy based on energy production from renewable sources.

### 3.1 Levels of PM2.5 in present and future scenarios

Figure 5 shows the PM2.5 mean annual concentration over Europe for the present period (1991-2010) and the changes projected in the future scenario (2031-2050, RCP8.5) both with emissions from ACCMIP and in the mitigation scenario (2031-2050,

REN80). As also stated in previous works (Tarín-Carrasco et al., 2019; Tarín-Carrasco et al., 2021), the chemistry/climate simulations revealed the presence of some areas in Europe exceeding the annual PM2.5 limit value (25 $\mu$g m$^{-3}$) established by the European Directive 2008/50/EC on ambient air quality and cleaner air for Europe. For the present period, Eastern Europe and some large cities and conurbations have the highest concentrations of fine particles, with some cities as Paris (France), Krakow (Poland) or Moscow (Russia) presenting elevated levels and exceeding 25 $\mu$g m$^{-3}$.

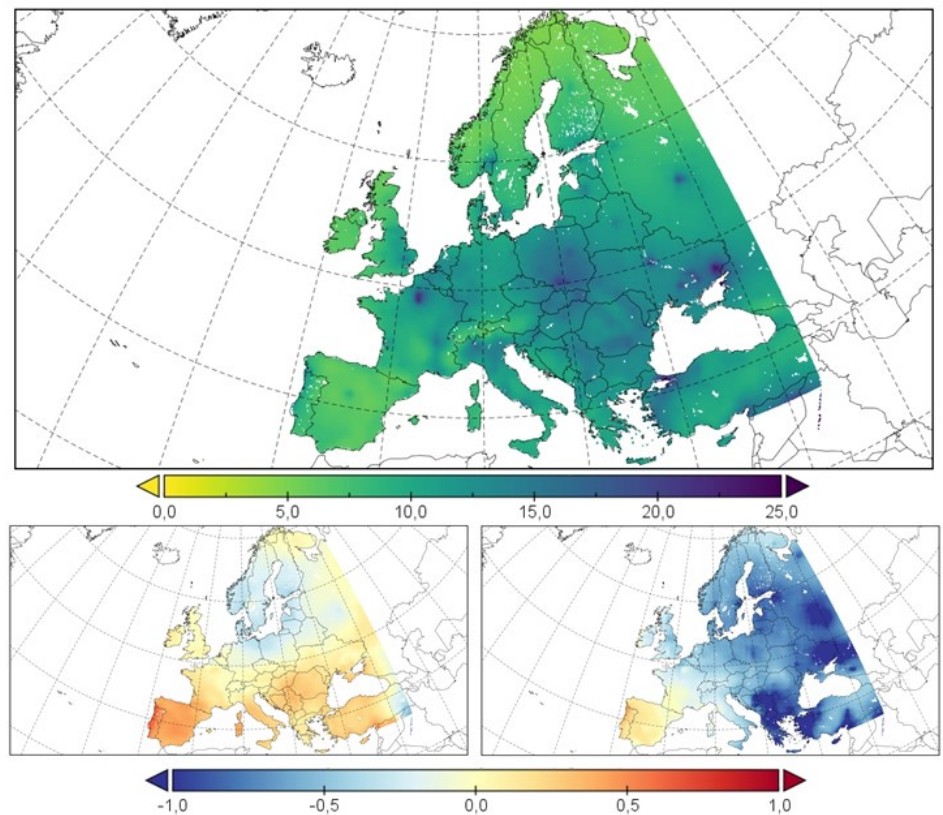

**Figure 5.** PM2.5 concentration over Europe for (top) present period (1991-2010); (bottom-left) difference with the future scenario (2031-2050, RCP8.5); and (bottom-right) difference with the future (2031-2050, RCP8.5)+80% renewables energies scenario (REN80). All units in $\mu$g m$^{-3}$.

The future changes in PM2.5 concentration attributable to GHG-induced climate change under the RCP8.5 scenario indicate a mean increase of 0.7 $\mu$g m$^{-3}$, with higher increases over southern of Europe. These results are in agreement with those of Silva et al. (2017) and Park et al. (2020), among others, who show an overall increase in surface PM2.5 concentration over most land regions. On the other hand, slight decreases are projected mainly over the Scandinavian countries and some Baltic areas (-0.2 $\mu$g m$^{-3}$). The decrease of PM2.5 over this area under the mitigation scenario (REN80) can be > 0.5 $\mu$g m$^{-3}$. Although

air quality improves overall in Europe; areas such as Paris, Krakow or Moscow will keep exceeding the European Directive threshold.

On the other hand, the REN80 scenario indicates an overall improvement of air pollution related to PM2.5 in Europe, especially over Eastern Europe. In that area, the effect of the REN80 largely counteracts the climate penalty (reductions of $>$-1.0 $\mu$g m$^{-3}$ with respect to present concentrations and $>$-1.4 $\mu$g m$^{-3}$ as average changes with respect to RCP8.5 scenario, reaching -2.5 $\mu$g m$^{-3}$ over certain hotspots). The reason for that decrease over Eastern Europe is the high ratio of fossil fuels in the energy mix of countries in that area (European Environment Agency, 2020). Again, those results are in agreement with previous works found in the scientific literature. For instance Liang et al. (2018) obtain a reduction of almost -0.9 $\mu$g m$^{-3}$ for a future scenario with a reduction of 20% in anthropogenic emissions due to power and industry sources.

It is also noticeable that northern Europe will benefit both from the climate penalty and the mitigation scenarios. This benefit cannot be observed over southern Europe, where the benefits of mitigation strategies might not compensate the effect caused by climate change on PM2.5 levels (fine particles will increase their concentrations in southern Europe as a consequence of decreasing precipitations and increased emissions from natural sources in this target area) (Jiménez-Guerrero et al., 2013b) (Figure 5).

### 3.2 Estimation of present premature deaths over Europe (PRE-P2010)

Figure 6 depicts the annual excess PD associated with present PM2.5 pollution (PRE-P2010). For the present period (1991-2010), 895,000 (95% CI 725,000-1,056,000) annual PD are estimated over the European area. This estimation is in agreement -although slightly higher- with that of Burnett et al. (2018) (647,000 PD), and exceeds that of Andersson et al. (2009) (546,000), who include only the contribution of primary PM2.5 and secondary inorganic aerosols. Crippa et al. (2019) report 260,000 PD associated to PM2.5, but the differences in the number of annual PD has to be sought primarily in the domain covered in the simulations presented here. In this sense, the target domain in this contribution covers the most populated areas or Russia or Turkey, which is not included in the numbers reported for Europe in the aforementioned study.

The highest incidence of excess PD are estimated in this contribution for Eastern Europe (467,000 annual PD, 95% CI 378,200-551,100) (Table 4), while the highest incidence of normalized PD per 100,000 adult inhabitants are found over Central Europe (229.4 PD/100,000 habitants per year). This latter result is in a good qualitative agreement with the estimation of Silva et al. (2016a), who identify the Benelux as the European region with the highest excess mortality rates associated to air pollution.

When considering individual endpoints (Figure 7), the main cause of excess PD associated to PM2.5 in the European domain is related to cardiovascular diseases (IHD+CEV), especially over Eastern Europe. Moreover, CEV and IHD are those causes of PD with the highest differences between each target area. The PD incidence over Eastern Europe is caused mainly by IHD+CEV, totalizing $>$ 60% of the premature mortality burden in Europe (Figure 7) because of (1) the higher population (51% of the total European population contemplated in this contribution) in comparison with Central (19% of European population) and Western Europe (30% of the European habitants); and (2) the higher PM2.5 concentrations (as previously shown in Figure 5). Those results are analogous for the total PD by all causes: PD are similar (with respect to their percentage) to the population

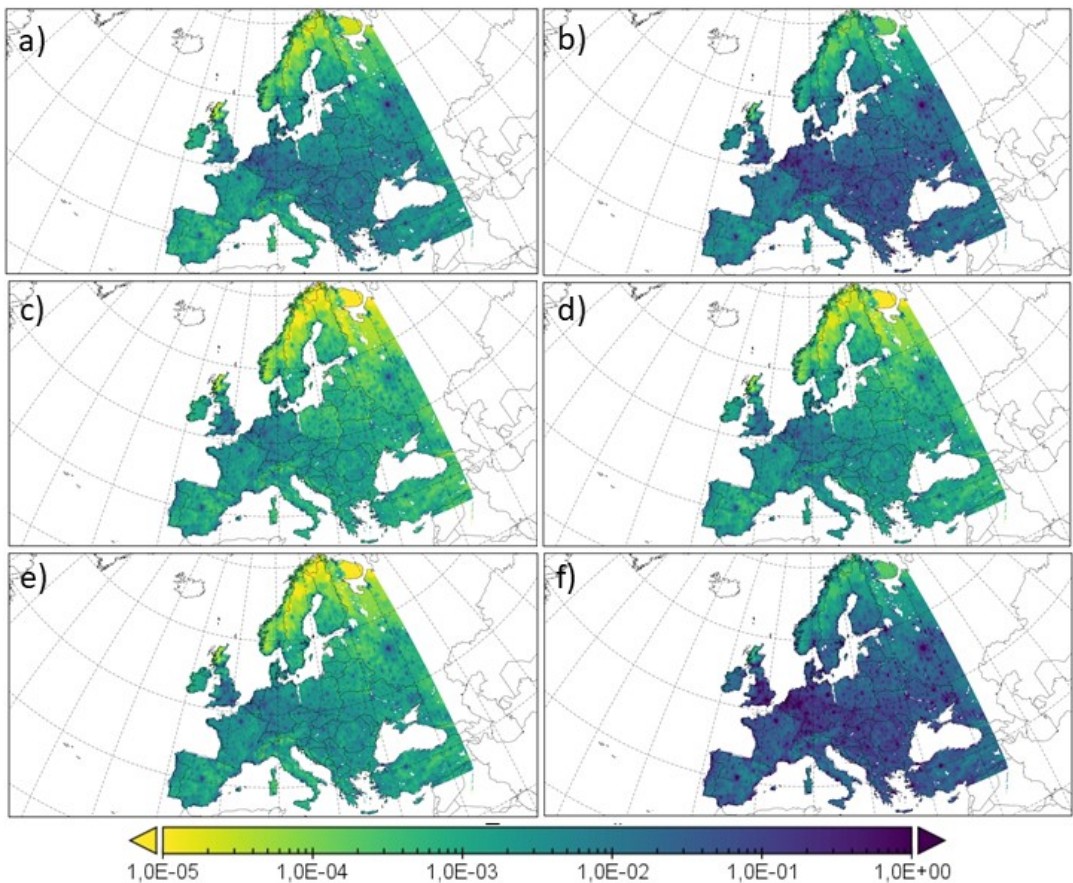

**Figure 6.** Estimation of PD over Europe for the PRE-P2010 case for different endpoints: (a) CEV, (b) IHD, (c) COPD, (d) LC, (e) LRI and (f) all. All units are PD/km$^2$ per year.

.

**Table 4.** Estimated incidence of excess premature deaths (PD, in thousands) and PD per 100,000 habitants for each European region in all scenarios covered. Numbers in parenthesis represent the 95% confidence interval.

| | Present | PRE-P2010 | RCP8.5 | FUT-P2010 | RCP8.5 | REN80-P2010 | RCP8.5 | FUT-P2050 | RCP8.5 | REN80-P2050 |
|---|---|---|---|---|---|---|---|---|---|---|
| | PD $\times 10^3$ | PD/100,000 h. | PD $\times 10^3$ | PD/100,000 h. | PD $\times 10^3$ | PD/100,000 h. | PD $\times 10^3$ | PD/100,000 h. | PD $\times 10^3$ | PD/100,000 h. |
| Western EU | 174.9 | 98.0 | 177.2 | 99.3 | 172.4 | 96.5 | 327.2 | 168.5 | 318.2 | 163.8 |
| | (141.7-206.4) | (79.4-115.6) | (143.5-209.1) | (80.4-117.2) | (139.6-203.4) | (78.2-113.9) | (265.0-386.1) | (136.5-198.8) | (257.7-375.5) | (132.7-193.3) |
| Central EU | 252.7 | 229.4 | 250.7 | 227.6 | 244.3 | 221.8 | 448.3 | 372.5 | 437.5 | 363.6 |
| | (204.7-298.2) | (185.8-270.7) | (203.1-295.8) | (184.4-268.6) | (197.9-288.3) | (179.7-261.7) | (363.0-528.9) | (301.7-439.6) | (354.4-516.3) | (294.5-429.0) |
| Eastern EU | 466.7 | 167.3 | 467.9 | 167.7 | 444.8 | 159.5 | 759.9 | 262.0 | 723.6 | 249.4 |
| | (378.0-550.7) | (135.5-197.4) | (379.0-552.1) | (135.8-197.9) | (360.3-524.9) | (129.2-188.2) | (615.9-896.7) | (212.2-309.2) | (586.1-853.8) | (202.0-294.3) |
| EUROPE | 894.3 | 167.5 | 895.8 | 157.8 | 861.5 | 151.8 | 1,535.4 | 253.9 | 1,479.3 | 244.7 |
| | (724.4-1,055.3) | (135.7-197.7) | (725.6-1,057.0) | (127.8-186.2) | (697.8-1,016.6) | (123.0-179.1) | (1,243.7-1,811.8) | (205.7-299.6) | (1,212.8-1,766.8) | (198.2-288.7) |

(PRE-P2010): PD for the present case; (FUT-P2010): PD for the future scenario with population at 2010 levels; (REN80-P2010): PD for the future mitigation scenario with population at 2010 levels; (FUT-P2050): PD for the future scenario with population projections of UN for 2050; (REN80-P2010): PD for the future mitigation scenario with population at 2010 levels; (REN80-P2050): PD for the future mitigation scenario with population projections of UN for 2050.

held in the respective areas. In this sense, Eastern Europe represents more than 50% of mortality in Europe, while Central and
Western Europe add around 20 and 30% of PD, proportional to their population.

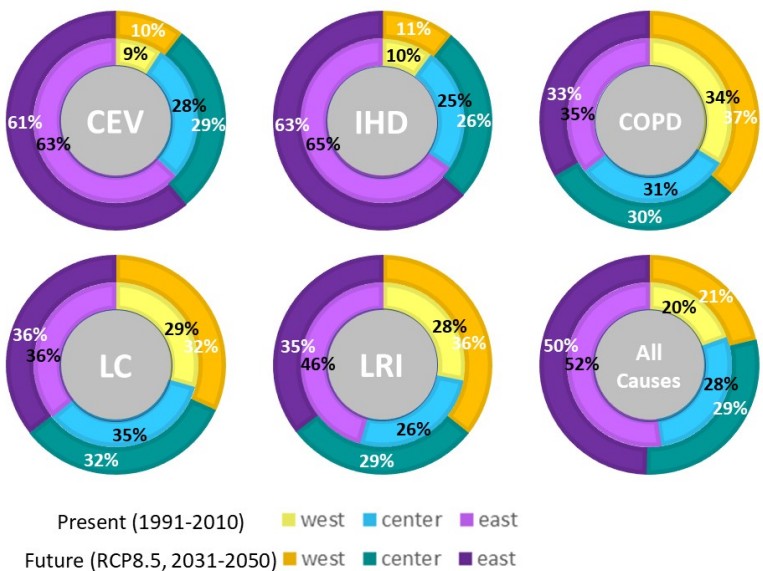

**Figure 7.** Relative contribution of each cause of PD by European region for the PRE-P2010 and the FUT-P2050.

.

It is noticeable that for COPD, LRI and LC, the three studied regions contribute with a similar percentage to total PD over
Europe (around 30%), which is indicative that the COPD, LRI and LC ratios are much higher over Western Europe (30% of
European population) and Central Europe (19% of total European population) than over Eastern Europe (51% of population).

### 3.3 Isolating the climate penalty effect over premature deaths over Europe (FUT-P2010)

In this Section the role of the climate penalty is assessed by comparing two simulations differing only in the radiative forcing
(1991-2010 vs. RCP8.5 2031-2050) (hence the two model runs FUT-P2010 vs. PRE-2010). For that, population has been
kept constant at 2010 levels. The results presented in Table 5 indicate that the climate penalty has a very limited impact on
air pollution-related future premature deaths over Europe. When comparing the total incidence of PD in the PRE-P2010 and
FUT-P2010 cases, only a +0.2% increase is projected in the future scenario. However, this increase is uneven among different
regions of Europe. While total PD per 100,000 adults (Table 4) increase in Western Europe by nearly 2% and in Eastern
Europe by 0.2%, the mortality decreases by -0.8% over Central Europe. These variations in the distribution of PM2.5 in
Europe, considering a future RCP8.5 scenario, is mainly due to changes in precipitation (Figure 8).

The decrease in rainfall in southern Europe (e.g. Jiménez-Guerrero et al. (2012); Domínguez-Morueco et al. (2019)) and
the increase in rainfall in northern and central Europe projected for this scenario (e.g. Jiménez-Guerrero et al. (2013b); Jacob

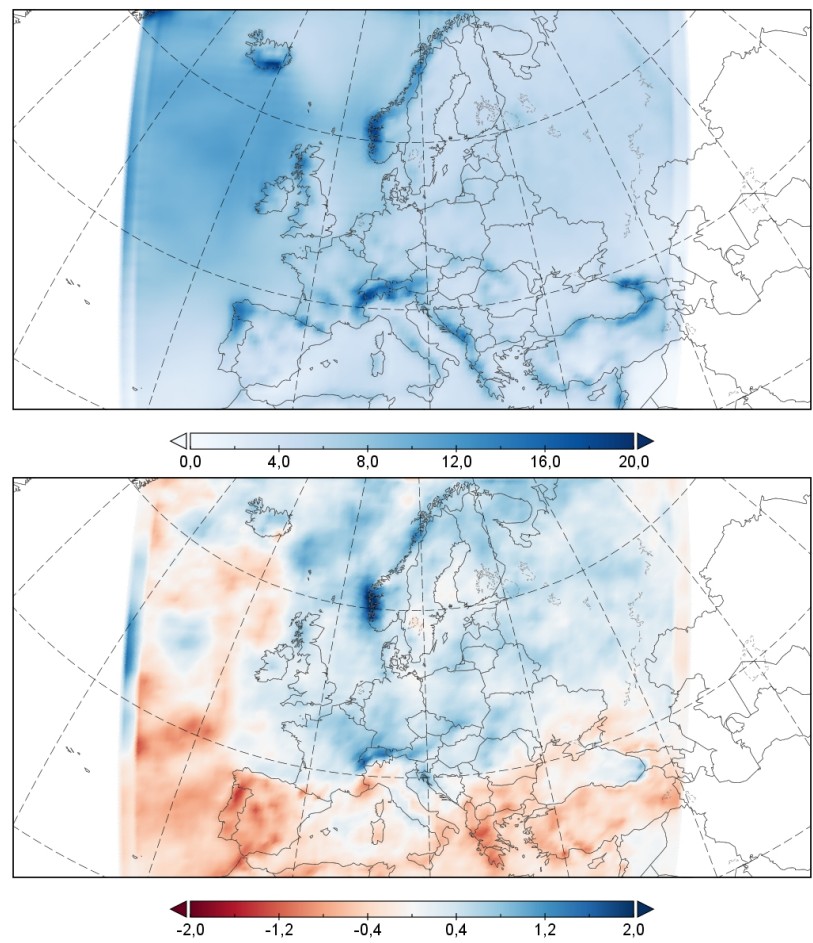

**Figure 8.** (Top) Present precipitation (mm day$^{-1}$) for the 1991-2010 period. (Bottom) Differences (mm day$^{-1}$) in future (RCP8.5, 2031-2050) minus present (1991-2010) precipitation.

et al. (2018)) affect the wet scavenging of particles (with a neglected influence of a modified photochemistry in the levels of particles). Wet scavenging represents, in this sense, the fundamental process for the removal of particles from the atmosphere (e.g. Ohata et al. (2016); de Bruine et al. (2018); Hou et al. (2018)).

It should be highlighted that the cause of premature mortality is most sensitive to changes in PM2.5 is CEV+IHD (Figure 9), as also found by previous studies (Pope et al., 2009, 2020). The spatial patterns of change of this latter endpoint condition the patterns of total variation of PD by all causes. IHD mortality rates increase by nearly 3% from 424,000 (95% CI 356,100-487,600) to 425,000 (95% CI 357,000-488,800) in the FUT-P2010 case. On the other hand, COPD, LC, and Other NCD barely change, since these causes are not too much sensitive to PM2.5 concentration as IHD and LRI at low PM2.5 concentrations (Figure 10), as also discussed in Tarín-Carrasco et al. (2021).

**Table 5.** Estimated annual premature deaths (PD $\times 10^3$) associated to fine particules for total population in all scenarios covered (in thousands).

| | Present | PRE-P2010 | RCP8.5 | FUT-P2010 | RCP8.5 | REN80-P2010 | RCP8.5 | FUT-P2050 | RCP8.5 | REN80-P2050 |
|---|---|---|---|---|---|---|---|---|---|---|
| | PD $\times 10^3$ | PD/100,000 h. | PD $\times 10^3$ | PD/100,000 h. | PD $\times 10^3$ | PD/100,000 h. | PD $\times 10^3$ | PD/100,000 h. | PD $\times 10^3$ | PD/100,000 h. |
| COPD | 27.7 | 4.9 | 27.8 | 4.9 | 26.5 | 4.7 | 52.2 | 8.6 | 49.8 | 8.2 |
| | (22.4-32.7) | (4.0-5.8) | (22.5-32.8) | (4.0-5.8) | (21.5-31.3) | (3.8-5.5) | (42.3-61.6) | (7.0-10.1) | (40.3-58.8) | (6.6-9.7) |
| LC | 47.6 | 74.7 | 47.7 | 74.8 | 45.4 | 71.8 | 68.7 | 121.7 | 65.3 | 116.8 |
| | (40.0-54.7) | (62.7-85.9) | (40.1-54.9) | (62.8-86.0) | (38.1-52.2) | (60.3-82.6) | (57.7-79.0) | (102.1-139.8) | (54.9-75.1) | (98.1-134.3) |
| LRI | 42.4 | 8.4 | 42.6 | 8.4 | 40.0 | 8.0 | 71.1 | 11.4 | 67.0 | 10.8 |
| | (35.6-48.8) | (7.1-9.7) | (35.8-49.0) | (7.1-9.7) | (33.6-46.0) | (6.7-9.2) | (59.7-81.8) | (9.6-13.1) | (56.3-77.1) | (9.1-12.4) |
| CEV | 86.9 | 15.3 | 87.1 | 15.3 | 81.8 | 14.4 | 151.6 | 25.1 | 142.4 | 23.5 |
| | (73.0-99.9) | (12.9-17.6) | (73.2-100.2) | (12.9-17.6) | (68.7-94.1) | (12.1-16.6) | (127.3-174.3) | (21.1-28.9) | (119.6-163.8) | (19.7-27.0) |
| IHD | 424.1 | 7.5 | 424.5 | 7.5 | 407.5 | 7.0 | 736.0 | 11.8 | 706.3 | 11.1 |
| | (356.2-487.7) | (6.3-8.6) | (356.5-488.1) | (6.3-8.6) | (342.3-468.6) | (5.9-8.1) | (618.2-846.4) | (9.9-13.6) | (593.3-812.2) | (9.3-12.8) |
| Other NCD | 265.6 | 46.8 | 266.2 | 46.9 | 260.3 | 45.9 | 455.8 | 75.8 | 448.5 | 74.2 |
| | (140.8-329.3) | (24.8-58.0) | (141.1-330.1) | (24.9-58.2) | (138.0-322.8) | (24.3-56.9) | (241.6-565.2) | (40.2-94.0) | (237.7-556.1) | (39.3-92.0) |
| All endpoints | 894.3 | 167.5 | 895.8 | 157.8 | 861.5 | 151.8 | 1,535.4 | 253.9 | 1,479.3 | 244.7 |
| | (724.4-1,055.3) | (135.7-197.7) | (725.6-1,057.0) | (127.8-186.2) | (697.8-1,016.6) | (123.0-179.1) | (1,243.7-1,811.8) | (205.7-299.6) | (1,212.8-1,766.8) | (198.2-288.7) |

(PRE-P2010): PD for the present case; (FUT-P2010): PD for the future scenario with population at 2010 levels; (REN80-P2010): PD for the future mitigation scenario with population at 2010 levels; (FUT-P2050): PD for the future scenario with population projections of UN for 2050; (REN80-P2010): PD for the future mitigation scenario with population at 2010 levels; (REN80-P2050): PD for the future mitigation scenario with population projections of UN for 2050.

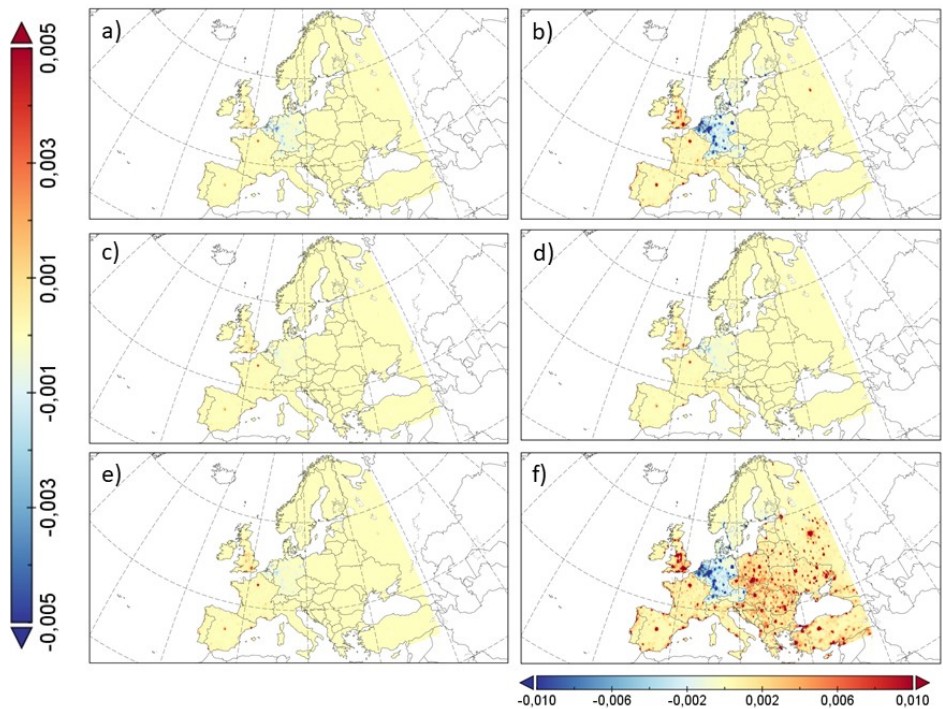

**Figure 9.** Differences between PD over Europe for the FUT-PRE2010 and the PRE-P2010 case for different endpoints: (a) CEV, (b) IHD, (c) COPD, (d) LC, (e) LRI and (f) all endpoints. All units are PD/km$^2$.

.

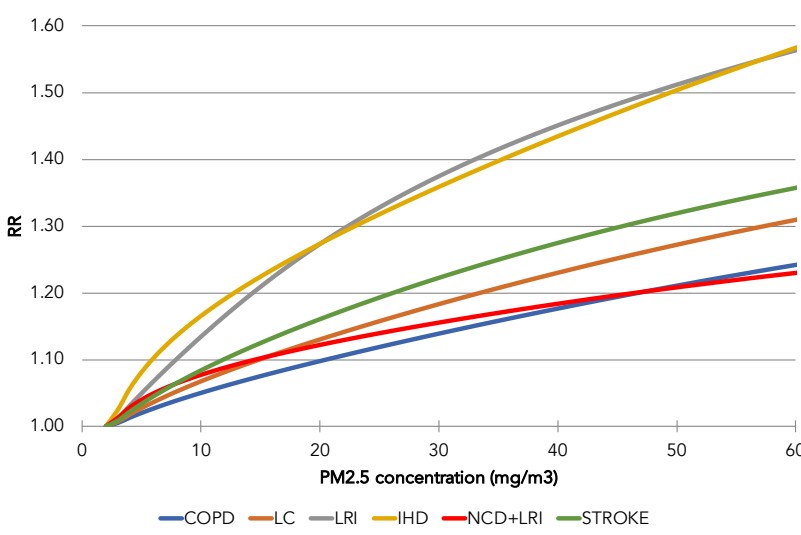

**Figure 10.** Shape of risk ratio functions for the target endpoints depending on PM2.5 concentration ($\mu$g m$^{-3}$).

### 3.4 Estimation of future premature deaths over Europe with a projected population (FUT-P2050)

The previous Section analyzed the contribution of climate change alone (RCP8.5 scenario, keeping population constant) to PD over Europe. Here, the impact of projected changes in the population following UN2050 data is added to climate penalty. This scenario is denoted as FUT-P2050. As shown in Figure 11 the highest mortality incidence in FUT-P2050 is related to IHD, which has an overspread spatial pattern in the target domain. For FUT-P2050, the projected PD incidence increases to 1,540,000 (95% CI 1,247,000-1,818,000), with Eastern Europe being the most affected region, with almost half of the excess

mortality rate over Europe (Table 4).

The premature mortality burdens associated with exposure to ambient PM2.5 in Europe are expected to increase by 72% in the year 2050 when comparing the PRE-P2010 and FUT-P2050 simulations, from 894,000 (95% CI 723,900-1,055,400) PD per year in 1990-2010 to 1,540,000 (95% CI 1,247,000-1,818,000) PD per year in 2031-2050 (Table 5). The leading cause of PM2.5-related mortality is IHD for both present and future cases: 424,000 (95% CI 356,200-487,800) PD per year, which

increases by 74%, to 736,000 (95% CI 618,200-846,400) PD per year in the FUT-P2050 case. IHD is followed by Other-NCD, with 265,600 (95% CI 140,800-329,300) PD per year increasing by 73%, to 458,400 (95% CI 243,000-568,400) PD per year in the FUT-P2050 case.

When assessing the relative contribution of each endpoint to the total PD burden both in PRE-P2010 and FUT-P2050, Figure 7 indicates that future climate change and the modification of population by the year 2050 will barely change the

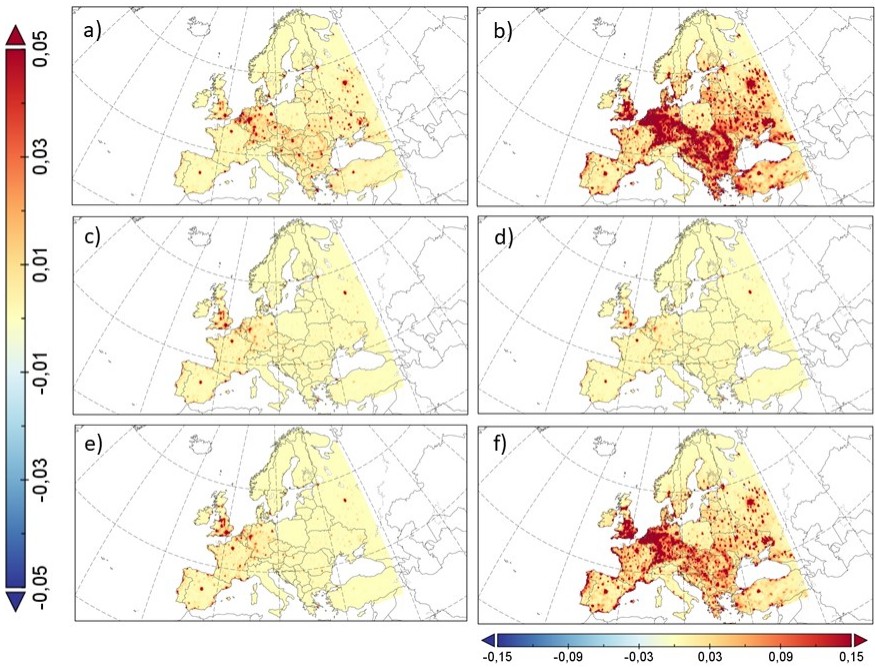

**Figure 11.** Differences in future premature mortality for the future scenario considering the population dynamics for 2050 (FUT-P2050). Endpoints considered are (a) CEV, (b) IHD, (c) COPD, (d) LC, (e) LRI and (f) All endpoints. All units in PD/km$^2$.

relative percentage of each mortality cause by region, so discussion is analogous to that presented in Section 3.2. The only exception is for LRI, which experiences several important changes between PRE-P2010 and FUT-P2050 (increasing from a 28% contribution in Western Europe for the present to 36% over this same area in the FUT-P2050 case; and from 35% in PRE-P2010 over Eastern Europa to 46% in FUT-P2050 over this same domain).

### 3.5   Effect of the future mitigation scenario (REN80-P2010 and REN80-P2050)

This case takes into account the climate change action (RCP8.5, 2031-2050) together with the emission scenario where it is assumed that 80% energy production over Europe is provided by renewables sources (REN80). For the analysis of this impact, a scenario where the population has been kept constant at 2010 levels (REN80-P2010) and a second scenario where the population dynamics has been taken into account (REN80-P2050) is analyzed below.

Figure 12 shows that in REN80-P2010 the PD incidence will be lower than in the REN80-P2050 scenario, which considers changes in the population, as previously discussed for the FUT-P2010 and FUT-P2050. Large central-Europe cities are the areas where the PD incidence will increase most with a changing population (increases in the density and age of population, as previously shown in Figure 1). A summary of the excess mortality incidence in all scenarios is presented in Table 4.

The incidence of all the mortality causes studied decreases in the future mitigation scenarios (REN80-P2010 and REN80-P2050) when compared to the corresponding FUT-P2010 and FUT-P2050 simulations, respectively, by 4%. This decrease

is especially found in central and eastern regions regarding the number of total PD (Table 4). This is explained by the fact
that an important part of the anthropogenic emissions of PM2.5 over Europe is associated to power generation (Crippa et al.,
2019). Precisely, this contribution is 15.1% here. Table 5 indicates that CEV and IHD present the largest relative reductions
of the incidence ratio of PD in the REN80-P2050 scenario with respect to FUT-P2050 (-6% changes in annual PD). COPD
and LRI show a -5% change between both scenarios, Figure 13), with around -4% cases less for the renewable scenario for
LC. Other NCD decreases by -2% in the REN80 simulations. Other NCD and CEV are the endpoints experiencing the largest
improvement in the mitigation scenario, with a decrease in annual excess PD of 30,000 (95% CI 15,900 - 37,200) and 10,000
(95% CI 5,300 - 12,400), respectively.

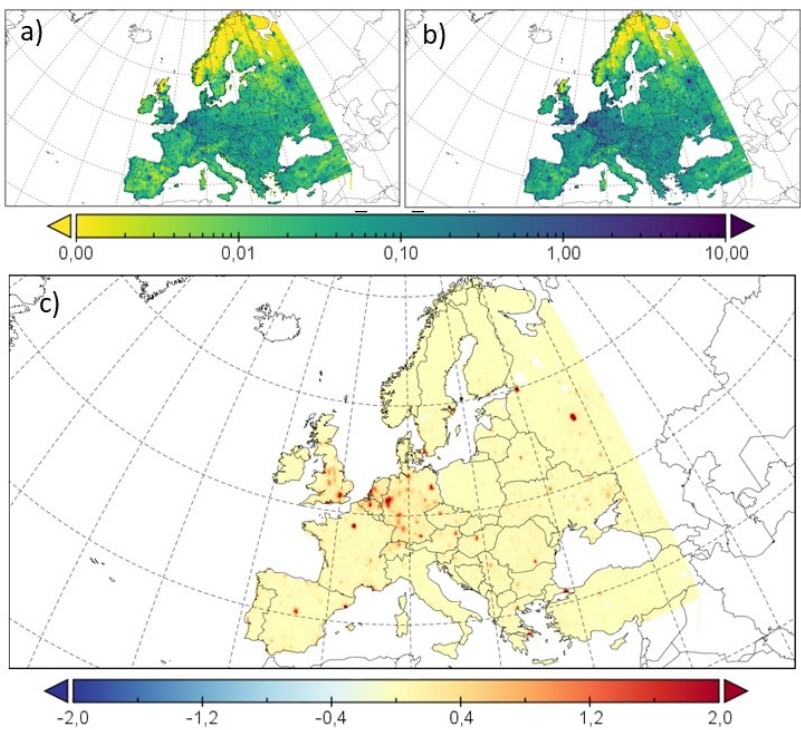

**Figure 12.** Future (2031-2050) premature deaths for (a) the mitigation scenario keeping constant the present population (REN80-P2010)
and (b) mitigation scenario considering the population projections for 2050 (REN80-P2050). (c) Differences between both future scenarios
(REN80-P2050 minus REN80-P2010). All units in PD/km$^2$.

### 3.5.1   Effect of the projected population for 2050

Last, this section tries to shed some light on the effect of the changes in the population projected (mainly, the aging of the
European population, since almost half all the PD over Europe occurs in the age group of 80+). A comparison of the mortality
burden for each age group considered among the different future scenarios is shown in the Supplementary Material (Table
SM4).

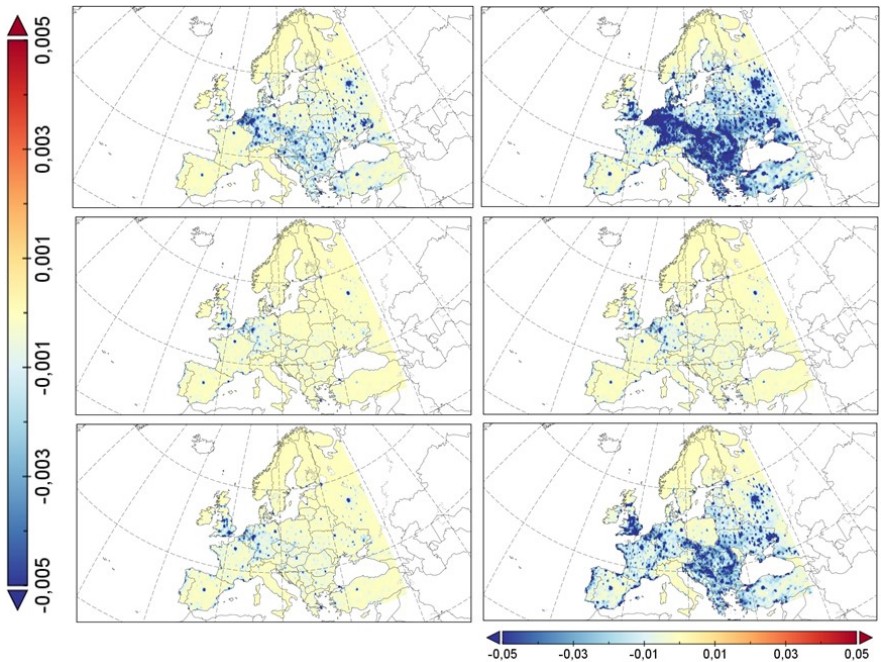

**Figure 13.** Differences in future premature mortality for the future scenario FUT-P2050 and the mitigation scenario (REN80-P2050), both considering the population projections for 2050. Endpoints considered are (a) CEV, (b) IHD, (c) COPD, (d) LC, (e) LRI and (f) All endpoints. All units in Premature Deaths (PD) per $km^2$.

In particular, the effect of the aging of population is clearly noticeable when comparing the FUT-P2010 and the FUT-P2050 scenarios, differing only in the population taking as basis for the estimation of PD over Europe. While the number of PD decreases in FUT-P2050 for younger age ranges, it generally increases for people over 65 years. For instance, the number of PD per 100,000 habitants in the 65-69 age range increases from 237 to 248 deaths/100,000 habitants; and the same statistics moves from 1037 to 1044 in the 80+ age range. This fact could be ascribed to the increase in the number of population falling within this group in the UN 2050 projection (Figure 1). Conversely, for younger age ranges, mortality will decrease in the future as population does. For instance, the 30-34 age range moves from 14 PD/100,000 habitants in the FUT-P2010 case to 13 PD/100,000 habitants in FUT-P2050. Overall, the total number of premature deaths will increase from 158 to 254 PD/100,000 (+61%) as a consequence of the aging of the population, despite a global decrease of -0.2% in total population over Europe is projected by the UN for the year 2050.

With respect to individual endpoints, the PD incidence caused by LRI keeps a similar ratio until the group 80+, when the incidence increases considerably (Figure 14). Hence, LRI is not strongly sensitive to the aging of population, but to the modifications in the number of dwellers. LRI increases from 42,000 (95% CI 35,200-48,300) in PRE-P2010 to 71,000 (95% CI 59,500-81,700) PD per year in the FUT-P2050 case. Also, LC results are to be highlighted (48,000 PD per year, 95% CI 40,300-55,200 in PRE-P2010 vs. 69,000 PD per year, 95% CI 57,900-79,400; associated to PM2.5 air pollution in the FUT-

P2050 case). For this cause, the number of deaths increases with the age up to a maximum for the age range 70-74. From there, the number of deaths decreases again. Finally, it is also noteworthy that PD in adults (25 until 60 years) decreases in the FUT-P2050 case for all the mortality causes studied. From 60 years and elder, PD increase in comparison with the present period due to the aging of the population in the future. For this reason, a higher incidence is estimated for aged people.

Hence, the differences between PRE-P2010 and FUT-P2010/P2050 must be sought in the differences among the risk ratios estimated by Equation 2. RRs are higher for younger age groups, with the incidence of PD higher for older groups. The baseline values ($y_0$) in Equation 2 are much higher for advanced ages, since elder people presents higher baseline mortality rates than younger dwellers.

## 4   Summary and Conclusions

This contribution has estimated the incidence of excess premature mortality associated to fine particulate matter for different present and future scenarios. The non-linear methodology employed depicted 894,000 (95% CI 723,900-1,055,400) annual premature deaths (PD) over Europe during the 1991-2010 period. The most important conclusions can be summarized as:

1. Effect of climate penalty: when the effect of climate penalty under the RCP8.5 scenario is isolated, the total premature mortality could be increased by around 2,000 PD (+0.2% in the FUT-P2010 vs. PRE-P2010 case), increasing from 894,000 (95% CI 723,900-1,055,400) to 896,000 (95% CI 725,500-1,057,800) over the target domain of Europe. Henceforth, the effect of climate penalty is limited due to the compensating effects of increased mortality over Western (+1.3 PD/100,000 h.) and Eastern Europe (+0.4 PD/100,000 h.), but a decrease over Central Europe (-2.2 PD/100,000 h.) (Figure 15) as a consequence of the reduced PM2.5 levels over this latter region under the RCP8.5 due to increased precipitation. In this sense, the domain of Central Europe (that includes northern and central areas of the continent) will benefit from the climatic effects, while the climate penalty will have an important effect in the domain of Western Europe, that includes southwestern European areas, where the concentrations of PM2.5 are projected to increase in the RCP8.5 scenario.

2. Impact of changes in the projected population: this modification, together with the climate penalty (FUT-P2050), leads to an excess premature mortality rate of 1,540,000 (95% CI 1,247,000-1,818,000), that is, an increase of 71.96% with respect to the present scenario and 71.67% to the future scenario in which only the climate penalty is considered. In contrast to the scenario in which population is kept constant (FUT-P2010), PD in central Europe in the future will increase because of the increase in the number of elderly population.

3. Impact of mitigation scenarios: The introduction of energy policies favoring renewable energies (80% of energy generated from renewable sources) could lead to a decrease of 60,000 (95% CI 48,600-70,900) annual PD in the year 2050 (1,480,000 PD per year, 95% CI 1,198,400-1,747,000, in the REN80-P2050 case, a decrease of -4%) in comparison with the FUT-P2050 case). So, mortality will importantly increase in the future scenario, but establishing mitigation policies based in renewable energies could reduce the number of PD, being Eastern Europe the most benefited area.

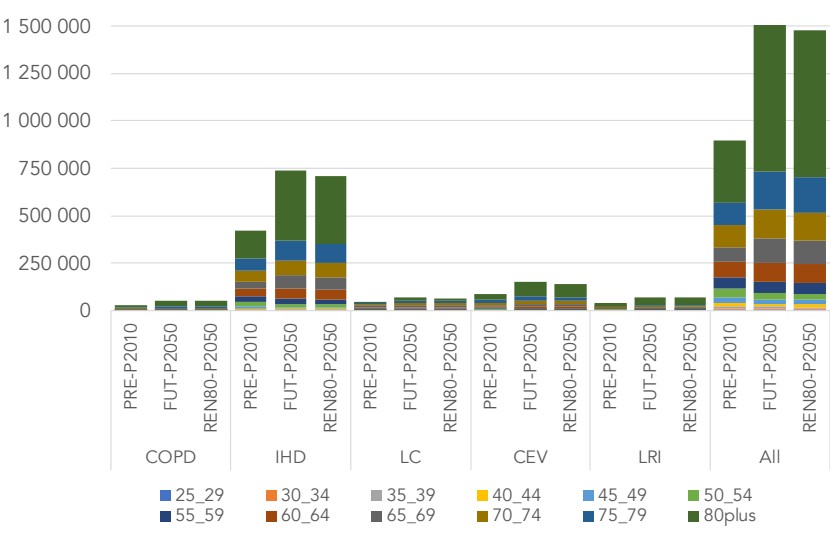

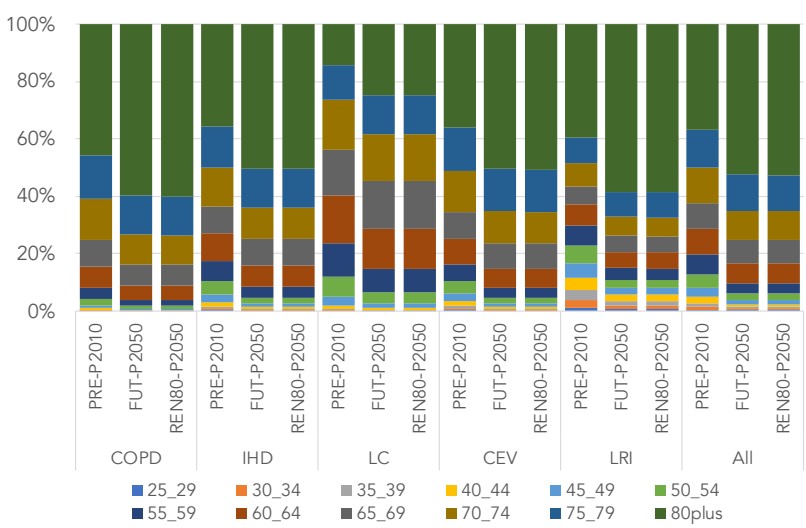

**Figure 14.** Present (PRE-P2010) and future (FUT-P2010/P2050) estimation of PD per pathology and age group over Europe (top, Total PD per year; bottom, %).

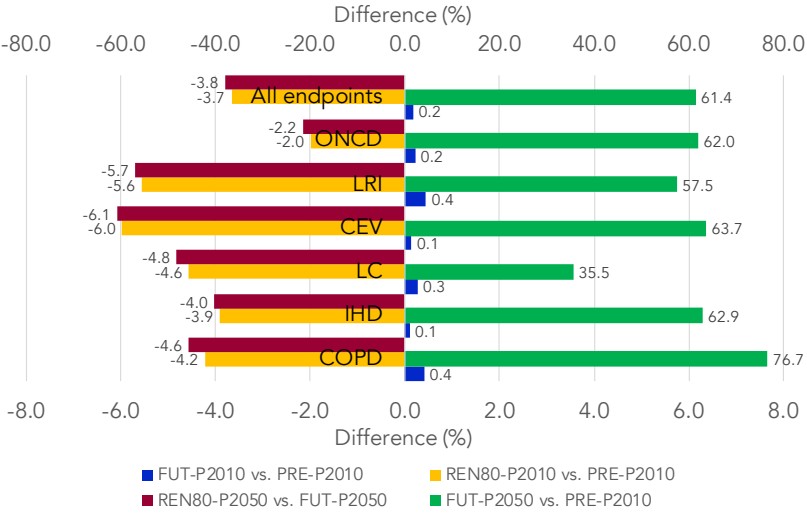

**Figure 15.** Relative differences for PD/100,000 habitants between the different scenarios for all the endpoints included in this work. FUT-P2010, FUT-P2050 and REN80-P2010 have been compared to the PRE-P2010 scenario, while the differences of the REN80-P2050 scenarios are estimated with respect to the FUT-P2050 scenario.

4. Causes of premature mortality: With respect to the different endpoints, IHD is the most important cause of mortality over Europe (over 50% of total premature mortality), with $\sim$ 424,000 (95% CI 356,200-487,800) annual PD in the present and the climate penalty scenario. When the change in 2050 population is considered, 736,000 (95% CI 618,200-846,400) annual PD are associated to this endpoint. This number reduces by 30,000 (95% CI 15,900 - 37,200) annual PD when the mitigation by renewables energy is considered. This results are caused by the high sensitivity of IHD to PM2.5. The second most important cause is CEV, with a $\sim$ 10% contribution to the total premature mortality. LRI, LC and COPD represent a 5%, 4% and 3%, respectively, of the total premature mortality. Last, it should be highlighted that other NCD causes represents a 30% of the PD in Europe.

This study presents an added value with respect to previous contributions. For instance, different baseline information has been included depending on the area of Europe considered. In addition, the PD incidence by age-range has been detailed here, highlighting the increase in future mortality not only by climate penalty, but because of an aged European population projected for the year 2050. The aging of the population will increase the number of sensitive dwellers and thus, the PD burden over the European domain.

Despite the robust results obtained, a number of issues should be addressed in further contributions. For instance, the difficulty to estimate biomass burning emissions under future scenarios has led to neglected emissions from this sector, that can be an important source especially when PM2.5 changes are less than 1.0 $\mu$g m$^{-3}$.

Furthermore, it is important to bear in mind that these results are estimated for different future cases and scenarios. Henceforth, these estimations are associated to different variables that are sometimes difficult to project, such as the death rate; the levels of air pollution (which depend on emission scenarios and the human influence on emissions like traffic, power generation or agriculture); and the demographic trends (elderly population, urbanization processes). Some of these variables are hard to control, and because of these difficulties the assumption of a constant baseline mortality rates ($y_0$) has been taken into account for estimating present and future premature deaths.

In addition, one of the aspects that deserve further attention with respect to this contribution is the election of the future forcing scenario (RCP8.5). This high-emissions scenario is frequently referred to as "business as usual" (BAU), suggesting that is a likely outcome if society does not make concerted efforts to cut greenhouse gas emissions (Pielke and Ritchie, 2021). This RCP8.5 was originally developed to represent an upper limit to climate impacts (Moss et al., 2010), and was intended to explore an unlikely high-risk future (Riahi et al., 2011). In this sense, several authors have highlighted that this worst-case scenario is an extremely-unlikely scenario and should not be treated as a BAU scenario consistent with high $CO_2$ forcing (Ritchie and Dowlatabadi, 2017; Ho et al., 2019; Peters and Hausfather, 2020). Despite the criticism on the election of RCP8.5 as a reference scenario (e.g. Grant et al. (2020); Peters and Hausfather (2020); Hausfather and Peters (2020); Pielke and Ritchie (2021)), other works keep the debate open. For example, Schwalm et al. (2020a, b) indicate RCP8.5, the most aggressive scenario in assumed fossil fuel use for global climate models, will continue to serve as a useful tool for quantifying physical climate risk, especially over near- to midterm policy-relevant time horizons. These same authors indicate that not only are the emissions consistent with RCP8.5 in close agreement with historical total cumulative $CO_2$ emissions (within 1%), but RCP8.5 is also the best match out to midcentury under current and stated policies with still highly plausible levels of $CO_2$ emissions in 2100. Other works, assuming RCP8.5 is not the most-likely scenario, point out that assessing RCP8.5 might be a helpful exercise, since it flags potential risks that emerge only at the extremes (O'Neill et al., 2016).

Despite the uncertainties and the limitations arising from this work (e.g. election of the forcing scenario, the use of a constant baseline mortality in all scenarios), it becomes clear from this contribution that governments and public entities must take project and clearly implement mitigation policies, which could improve air quality and therefore, the wellness of the European citizens.

*Data availability.* The modeling data are available on the Zenodo repository (doi:10.5281/zenodo.6230393). For further information, please contact the corresponding author: pedro.jimenezguerrero@um.es.

*Author contributions.* PT-C wrote the manuscript, with contributions from PJ-G. LP-P and PJ-G designed the air quality/climate experiments; P-TC conducted the numerical estimations of premature mortality, with the support of UI, PJ-G and CG. P-TC did the analysis, with
the support of UI, CG, LP-P and PJ-G.

*Competing interests.* The authors declare no conflict of interest.

*Acknowledgements.* The authors are thankful to the WRF-Chem development community and the G-MAR research group at the University of Murcia for the fruitful scientific discussions.

*Financial support.* The authors acknowledge the ACEX project (CGL2017-87921-R) of the Ministerio de Economía y Competitividad/Agencia
Estatal de Investigación of Spain, the ECCE project (PID2020-115693RB-I00) of Ministerio de Ciencia e Innovación/Agencia Estatal de Investigación (MCIN/AEI/10.13039/501100011033/) and the European Regional Development Fund (ERDF/FEDER Una manera de hacer Europa). This work has also received funding from the European Union's Horizon 2020 research and innovation programme under grant agreement No 820655 (EXHAUSTION). L. P.-P. was supported by the FPU14/05505 grant from the Spanish Ministry of Education, Culture and Sports.

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
