# Peer review of "Reducing future air pollution-related premature mortality over Europe by mitigating emissions from the energy sector: assessing an 80% renewable energies scenario"

_Atmospheric Chemistry and Physics, 2021_

## Author Comment (AC2)

**Response to Reviewers: Reducing future air pollution-related premature mortality over Europe by mitigating emissions: assessing an 80% renewable energies scenario**

**Reviewer ♯1**

The authors have investigated premature mortality changes due to PM2.5 over Europe under present and future conditions. [...] It is an interesting paper that separates different causes of premature death including climate changes, emission changes due to mitigation to renewable energy, and population changes.

A: We would like to acknowledge Reviewer ♯1 for the very positive view on the manuscript and fruitful comments. Please find below our item-by-item response to the concerns raised.

**Major comments**

1. The authors said their results are in agreement with the results of Andersson et al. (2009) and Crippa et al. (2019) without giving any numbers of previous studies (line 211). The authors estimated premature deaths due to PM2.5 over Europe to be 895,000, which is a lot higher than 546,000 of Andersson et al. (2009) and 210,000 of Crippa et al. (2019). As referee ♯2 pointed out, this value is also higher than 647,000 of Burnett et al. (2018) with the same GEMM method. Values from those three previous studies are even below 725,000 which is the lower end value of this study in terms of 95% confidence interval. The authors should give the reason or justify their premature death calculations.

   A: The reviewer is right. The difference in the estimated premature deaths comes mainly from the domain covered in our simulations. In this sense, the domain covers the most populated areas or Russia or Turkey, which were not included in the studies of Andersson et al. (2009), Burnett et al. (2018) or Crippa et al. (2019). The statement that indicates that our results are in agreement with those of previous works (despite several methodological differences and baseline periods) refer to the same area of study. This has been clarified in the revised version of the manuscript.

2. Following up on the comment ♯1, the reason could be due to the old aerosol scheme (GOCART) used in this study, which was developed 20 years ago (Table SM1). [...] The model overestimated the observed AOD over Europe (Figure 2 in Palacios-Pena et al., 2020), and I speculate the surface PM2.5 might also be overestimated similarly in the model, which made higher premature deaths of this study compared to previous studies (comment ♯1). However, premature deaths are calculated by PM2.5, not AOD. The authors should evaluate their model against the observed surface PM2.5, especially when using the 20-years old aerosol scheme.

   A: The model has been previously evaluated against ground-based observations of PM2.5 in a number of works (e.g. Jerez et al., 2013; Ratola and Jiménez-Guerrero, 2016, Im et al., 2018). These results indicate a slight tendency of the model to underestimate the levels of PM2.5 over the target domain, despite AOD is slightly overestimated (as indicated by the Reviewer). Part of this discrepancy could be ascribed to the methodology for estimating AOD in the work of Palacios-Pena et al. (2020), which relies on Malm et al. (1994). Therefore, the higher number of premature deaths estimated on this contribution cannot be related to the overestimation of ground-level PM2.5. We agree with the reviewer that this information is useful in the text. Hence, instead of including a full evaluation of the PM2.5, the following paragraph has been included in the revised version of the manuscript:

   "The parameterizations implemented in the WRF-Chem model are summarized in Table SM1 of the Supplementary Material. The robustness of this simulation for representing particulate matter or its chemical components was evaluated in other works, which rely on the same data and include the assessment of the integrations (e.g. Jerez et al. (2020); Palacios-Peña et al. (2020); Pravia-Sarabia et al. (2020); Jerez et al. (2021); López-Romero et al. (2021), among others). For instance, the validation of the modelling system used revealed negligible errors for PM2.5 over large areas of Europe but a tendency underestimation, especially over northern Germany and the central and western Mediterranean (around -0.2 to -0.4 $\mu g\ m^{-3}$) when compared to stations belonging to the EMEP air sampling network (Jerez et al., 2013). In addition, Ratola and Jiménez-Guerrero (2016) found that the modelling

45     system provided consistently mean fractional biases (MFB) and mean fractional errors (MFE) in the range of 20-30% and 30-60%, respectively, when evaluating a PM2.5 component (BaP) against against EMEP stations in Europe. These results support the use of this modelling system as a good depiction of PM2.5 air concentrations and as a valid tool to provide information for health studies, as those presented in Tarín-Carrasco et al. (2019); Tarín-Carrasco et al. (2021), which rely on these very simulations."

3. It is unclear whether the authors included biomass burning emissions in the simulation. I think biomass burning contributes relatively a small fraction of PM2.5 mass over Europe ( 10-20%), but it can be an important source especially when PM2.5 changes are less than 1.0 $\mu$g m$^{-3}$ (Figure 4). Because RR in eq (1) has a non-linear response to PM2.5 concentration, the correct representation of biomass burning can be also important to accurately calculate premature death.

A: The reviewer raises an interesting point. However, biomass burning emissions in the simulations are derived from a climatology and have been neglected in our simulations, due to the difficulties to estimate future biomass burning emissions. This is a limitation of this work with has been discussed in the conclusions.

4. The authors did a good job in calculating emissions for 80% renewable energy scenario (REN80). However, in reality, there will be reductions in emissions from other sectors as well, similar to what we observed from 1990 - 2017 trends over Europe (https://www.eea.europa.eu/data-and-maps/indicators/main-anthropogenic-air-pollutant-emissions/assessment-6). One example could be vehicle emissions, emission factors of recent vehicles (EURO-4 or EURO-5) were significantly decreased compared to old vehicles (EURO-1 or EURO-2) (Figure 4 in Huang et al., 2018). This means that baseline PM2.5 will be different in the future. Not only differences in PM2.5 but also baseline PM2.5 is important in calculating changes in premature death, as the RR function is non-linear. For example, PM2.5 changes of 15 -> 10 $\mu$g m$^{-3}$ would give different premature death changes compared to PM2.5 changes of 10 $\mu$g m$^{-3}$ -> 5 $\mu$g m$^{-3}$.

A: We strongly appreciate the valuable comment of Reviewer ♯1. We fully agree with the reviewer. The strong non-linearity of the RR function may lead to important differences in future scenario. However, this contribution intends to provide a sensitivity analysis of the impacts of renewable energies on the power-generation section. Changing baseline PM2.5 emissions in the future (for instante, introducing projected changes in EURO-x emissions in vehicles) does not allow to isolate the impact of mitigation policies aimed at enhancing the implementation of renewable energies in the power generation sector. Therefore, despite we fully agree with the reviewer in the sense that a very interesting study would be to project future baseline emissions. However, this is beyond the scope of this contribution.

5. In Section 3.3, the authors explain the main reason for PM2.5 changes is precipitation and wet scavenging. This is also conclusion ♯1 of their paper. This is an important finding that climate can cancel/enhance PM2.5 changes in a specific region of Europe. However, no figures were presented in the manuscript. Please consider a more detailed discussion on changes in precipitation and resulting PM2.5 with figures (wet deposition fields would be best) if possible. And does the cloud also affects PM2.5 through changes in photochemistry, or only precipitation is important?

Despite the decrease in precipitation (and consequent enhancement of the wet scavenging) is documented in the text (Section 3.3), we have included a new Figure of precipitation changes in this section (wet scavenging was unfortunately not stored in the database), as suggested by the Reviewer. As indicated in the cited works, precipitation leads the modifications of PM2.5. This has been clarified in the revised version of the manuscript.

**Minor comments**

1. Section 2.1. Chemistry models typically simulate dry aerosol mass, but I think aerosol liquid water should also be included in mortality calculation. How did the authors calculate aerosol liquid water in addition to dry aerosol mass?

A: The reviewer is right, and models typically simulate dry aerosol mass, and it is this value which is used by the cited works in this contribution (Silva et al., 2016; Lelieveld et al., 2013, 2015, 2019; Tarín-Carrasco et al., 2021, among many

others). Therefore, it is PM2.5 dry aerosol mass which is used for the estimations presented here. This has been clarified in the revised version of the manuscript.

2. Table 2 nicely shows emissions from the energy sector for two scenarios. Please consider adding anthropogenic emissions from other sectors in another column, so that readers can see the effects of REN80 on total anthropogenic emissions over Europe. The Sum of all other sectors could be OK, although the emission from each sector would be preferred.
A: We agree with the reviewer's suggestion. This information has been included in the revised version of the manuscript.

3. Line 207: What are the natural sources the authors referring to? Does it biogenic VOCs? biomass burning? dust? or sea salt?
A: Natural sources refer to dust, sea salt and biogenic VOCs. Following the reviewer's suggestion, we have clarified it in the revised version of the manuscript.

4. Line 215: Please provide the quantitative numbers from the previous study.
A: In the cited Silva et al.'s work, there is not an exactly quantitative assessment, but a clear identification of the Benelux as the European region with the highest excess mortality rates associated to air pollution. This has been clarified in the the reviewed version of the manuscript.

5. Line 245: Can authors provide RR function shape to show sensitivities of different disease categories to changes in PM2.5?
A: A graphic including the RR function shape has been included in the revised version of the manuscript.

6. Figures 7 and 10: In some panels (a, c, d, e), I can't see differences due to the color scale. It looks like the color scale is adjusted for panel f). Please consider adding a colorbar under each panel with a different color scale.
A: For the sake of clarity in the comparison of the different panels and figures, we humbly believe that keeping the same colorbar for all the panels can help interpreting the importance of each endpoint and their comparability. However, for the sake of clarity, and following the reviewer's advice, we have used a different colorbar for the total endpoints (panel f).

110

**General:** Tarin-Carrasco et al. use WRF-Chem to study emissions scenarios and subsequent impacts on human-health through-out Europe. [...] I outline my comments below and attempt to provide guidance, where possible.

A: We would like to acknowledge Reviewer ♯2 for the very positive view on the manuscript and fruitful comments. Please find below our item-by-item response to the concerns raised.

115

**Specific/Major Comments:**

– The REN80 scenario is framed as an "80% renewable energies scenario." However, this scenario only considers emissions from the energy sector. Meanwhile, mobile, residential, and industrial emissions are left alone. This seems to be an important oversight since these other sectors have clear impacts on the trajectory of the energy sector. For example,

120  vehicular electrification. If adoption of electric vehicles expands, as is expected, this will have a large effect on the energy sector. If residential heaters using wood combustion are replaced with electric heaters, that too would have a massive impact on primary PM emissions. It seems the title of the manuscript does not quite match the contents. [...]

A: We agree with the comments raised by the reviewer. In this sense, the title of the manuscript has been changed as suggested by the Reviewer to "Reducing future air pollution-related premature mortality over Europe by mitigating

125  emissions from the energy sector: assessing an 80% renewable energies scenario"

In addition, what's notable to me is that we are today (i.e. 2021) closer to the "Future" time slice (only 10 years away from 2031) than the "Present" time slice (11 years away from 2010). [...] Why are the older RCP scenarios used and not the newer SSP scenarios? Why is the RCP 8.5 scenario used when it is such an unlikely scenario (Hausfather and Peters, 2020)? If one of the more "likely" scenarios were used, perhaps much of the reported climate penalty in the Iberian

130  Peninsula would diminish.

A: The reviewer raises an interesting point. As commented in the manuscript, the results derive from the REPAIR initiative, sent for evaluation in 2015 and funded later by the Spanish Ministry. That project aimed to provide a large database of different scenarios and model set-ups (over 500 years of simulations). At the moment the simulations started, there was not information provided by the newer SSP runs so RCP scenarios were implemented. At the same time, the

135  reference period chosen agreed with that selected by the IPCC in their AR5 for some studies (1991-2010 and 2031-2050), and RCP8.5 was a feasible scenario (despite it represented the top of the forcings included in the RCPs). Because of that, and the large time needed for regional air quality-climate simulations (over three years), the simulations included here used the RCP8.5 scenario. We are aware that that might be a limitation of the study, and hence this is discussed in the Conclusions section.

140  – Also, please indicate in Table 3 which year of ACCMIP emissions were used for each case. I believe the FUT-P2010/50 simulations used present day emissions from ACCMIP, but it isn't clear to me.

A: ACCMIP emissions refer to the year 2000. The reviewer is right: FUT-P2010/50 simulations use present-day ACCMIP emissions. This has been clarified in the revised version of the manuscript.

– Health impact assessments associated with air pollution exposure traditionally report results in terms of premature deaths.

145  Premature deaths are a tangible number that is easy to understand (unlike YLLs and DALYs, in my opinion). However, issues can arise when attempting to communicate health burdens attributable to air pollution exposure. [...] I believe the authors have an opportunity to (and should) update the presentation of their results in a way that can help the community more broadly communicate the health burden associated with air pollution exposure. [...] I believe health impacts reported in terms of decrements of life expectancy can mitigate the influence of non-air pollution exposure effects. [...]

150  A: We agree with the reviewer's comments. We are aware that presenting health impacts associated to premature deaths, the results can be skewed by changes in the size of the underlying population and age structure. We discussed largely among the authors on how to present the results, and we humbly believe that the pros in terms of social communication

of presenting the information of premature deaths per 100,000 habitants (despite they can be conditioned by the age structure) are larger that the cons related to the difficulty to interpret other parameters as years of life lost, or decrements of life expectancy. So, we humbly believe that results are easy to interpret and analyze the way they are presented in the manuscript (both in terms of premature deaths and deaths per 100,000 habitants).

– The present-day health impact numbers reported here are noticeably high: 895,000 for Europe. For example, Burnett et al. 2018 report 647,000 premature deaths for Europe. Does the WRF-Chem set-up used here have a high PM bias? I'm curious, why not use "observed" PM from one of the various satellite products and apply the change in PM calculated from the CTM to one of those datasets? Essentially, use the CTM to calculate the sensitivity of each scenario and apply to "observations."

A: The results of the model validation, showing a good skill of the model for representing PM2.5, are now included in the revised version of the manuscript (please see the Answer to Question 2 of Reviewer $\sharp$1).

– The differences in endpoint are discussed throughout the manuscript, including how the proportions change over time (e.g. much of Sections 3.5.1, 3.4, and Fig. 6). However, I believe the same baseline mortality rates ($y_0$ in Eqn. 1) are used in both "present" and "future" simulations. It seems that these comparisons are difficult to make under those assumptions.

A: The reviewer is right. $y_0$ refer to the same in both period because of extreme difficulty to provide an accurate estimation of future $y_0$ due to its dependency of factors as the economy or sanitary aspects on the countries. We are well aware that this is a limitation of this type of studies, and as that it has been highlighted in the Conclusion section.

**Technical/General Comments:**

– "Worldwide air quality has worsened in the last decades as a consequence of increased anthropogenic emissions, in particular from the sector of power generation." Worsening air quality has certainly been the case in some locations, but far from all. For example, air quality in China, home to nearly $\approx$20% of the global population, has dropped precipitously in the last decade.

A: We fully agree with the reviewer's comment. The sentence has been rephrased.

– An 80% renewables adoption only yields a reduction in PD of 4%. That is a surprising result and likely because the 80% scenario only considers the power generation source of emissions in a present-day environment.

A: We strongly appreciate the valuable comment of Reviewer $\sharp$2. This contribution intends to provide a sensitivity analysis of the impacts of renewable energies on the power-generation section. Changing baseline PM2.5 emissions in the future including changes in other sectors does not allow to isolate the impact of mitigation policies aimed at enhancing the implementation of renewable energies in the power generation sector, which is precisely the aim of this contribution.

– Line 34: exposition > exposure
A: This has been corrected in the revised version of the manuscript.

– Line 36: define "this pollutant"
A: This has been defined in the revised version of the manuscript.

– Line 37: stablished > established
A: This has been corrected in the revised version of the manuscript.

– Line 41: As constructed, this sentence is really confusing. First, the respiratory impacts that are cited are for ozone, not PM. Second, why cite global numbers when that table has European numbers and that is the focus of the effort here?
A: This has been corrected in the revised version of the manuscript.

- Line 44: As constructed, this 70% number is incorrect. 70% of global mortality (i.e. total mortality) is not attributable to PM. That would be something like 45-50 million premature deaths a year. Do the authors mean 70% of air quality attributed mortalities are attributable to PM? I do not know, please re-visit that citation.

  A: This has been clarified and corrected in the revised version of the manuscript.

- Line 57: Please revise the writing of "In Europe, despite agriculture is the sector": it is confusing as written.

  A: This has been corrected in the revised version of the manuscript.

- For each case considered, are the 20-years of simulation simply averaged together? This should be noted in Section 2.3.

  A: Estimations are calculated for each year and then averaged for the 20-year period. A statement clarifying this has been added in the revised version of the manuscript.

- Line 161: The PM decrease is strictly a primary PM decrease. I suspect secondary PM to be the dominant source of PM exposure. With that said, how is this primary PM SPECIATED and are there volatility assumptions for the organic portion?

  A: PM speciation and volatility assumptions used can be found in Im et al. (2015). Evaluation of operational online-coupled regional air quality models over Europe and North America in the context of AQMEII phase 2. Part II: Particulate matter. Atmos. Environ. 115, 421-441. This reference and the information within has been introduced in the revised version of the manuscript.

- Table 3: Indicate which year of ACCMIP emissions were used for each case.

  A: ACCMIP emissions refer to the year 2000. This has been introduced in the revised version of the manuscript.

- Line 188: "cities" written twice.

  A: This has been corrected in the revised version of the manuscript.

- Line 194: Do the authors mean "decrease" in the REN80 scenario? I see a decrease in Fig. 4 over the Baltic states.

  A: We appreciate the reviewer's observation. The reviewer is right, and that has been corrected in the revised version of the manuscript.

- Line 212: Crippa et al. (2019) seems to indicate around 260,000 for Europe, far less than 895,000.

  A: As indicated previously, the numbers reported by Crippa et al. (2019) leave out of the target domain important areas as Turkey or Russia, which have been included in our contribution. In addition, the numbers obtained by Crippa et al. (2019) are in the lowest range of the scientific literature included in this contribution (e.g. 647,000 premature deaths estimated by Burnett et al., 2018). The difference in these numbers has been detailed in the revised version of the manuscript.

- Something seems strange in Fig. 7. For example, Poland. It essentially has no shading in panels a)-e). However, it is full of red in panel f). Same throughout the Balkans and Eastern Europe. Is panel f) not the summation of panels a)-e), because this figure seems to say no.

  A: For the sake of clarity in the comparison of the different panels and figures, we humbly believe that keeping the same colorbar for all the panels can help interpreting the importance of each endpoint and their comparability. However, for the sake of clarity, and following the reviewer's advice, we have used a different colorbar for the total endpoints (panel f).

- Section 3.5: Several times, "FUT-2050" was used. That should be "FUT-P2050". There is a lot of repetition in Sections 3.4 and 3.5.1.

  A: We appreciate the reviewer's observation. This has been corrected in the revised version of the manuscript.

- Figure 10: A "P2050" case is mentioned in the figure description. Please keep all scenario names consistent (i.e. only use Table 3 names).

  A: This has been corrected in the revised version of the manuscript.

– Line 308: These few sentences on lung cancer are highly speculative. I recommend deleting them.

235     A: These sentences have been deleted in the revised version of the manuscript.

– I recommend getting rid of the log-scales on Fig. 11. That presentation tends to exaggerate the age bins that have minor impacts (younger ages) and deflate the age bins where most impacts occur (older ages).
A: Figure 11 has been modified in the revised version of the manuscript.

– Line 339: "trusting" is a strange word to use here.

240     A: This has been changed in the revised version of the manuscript: "So, mortality will importantly increase in the future scenario, but establishing mitigation policies based in renewable energies could reduce the number of PD, being easter Europe the most benefited area."

– Line 340: easter > Eastern
A: This has been corrected in the revised version of the manuscript.

---

## Author Response (AR2)

**Response to Reviewers: Reducing future air pollution-related premature mortality over Europe by mitigating emissions: assessing an 80% renewable energies scenario**

**Reviewer ♯1**

The authors have addressed most of the comments raised by reviewers. However, there are still a couple of points that were not addressed and must be addressed before the publication.

A: We would like to acknowledge Reviewer ♯1 for the very positive view on the manuscript and fruitful comments. Please find below our item-by-item response to the concerns raised.

**Major**

1. RCP8.5 scenario: The authors presented practical reasons for the RCP8.5 scenario (e.g., REPAIR initiative, simulation time), but did not provide the scientific justification of the use of RCP8.5. In their response, there was no discussion about the paper the reviewer had raised (Hausfather and Peters, 2020). They said it was discussed in the Conclusions section but it was not clear - why an unlikely scenario was chosen and why RCP8.5 was still feasible for this study.

   A: The reviewer is right. We have included the following discussion at the end of the conclusion section:

   *In addition, one of the aspects that deserve further attention with respect to this contribution is the election of the future forcing scenario (RCP8.5). This high-emissions scenario is frequently referred to as "business as usual" (BAU), suggesting that is a likely outcome if society does not make concerted efforts to cut greenhouse gas emissions (Pielke and Ritchie, 2021). This RCP8.5 was originally developed to represent an upper limit to climate impacts (Moss et al., 2010), and was intended to explore an unlikely high-risk future (Riahi et al., 2011). In this sense, several authors have highlighted that this worst-case scenario is an extremely-unlikely scenario and should not be treated as a BAU scenario consistent with high $CO_2$ forcing (Ritchie and Dowlatabadi, 2017; Ho et al., 2019; Peters and Hausfather, 2020). Despite the criticism on the election of RCP8.5 as a reference scenario (e.g. Grant et al. (2020); Peters and Hausfather (2020); Hausfather and Peters (2020); Pielke and Ritchie (2021)), other works keep the debate open. For example, Schwalm et al. (2020a,b) indicate RCP8.5, the most aggressive scenario in assumed fossil fuel use for global climate models, will continue to serve as a useful tool for quantifying physical climate risk, especially over near- to midterm policy-relevant time horizons. These same authors indicate that not only are the emissions consistent with RCP8.5 in close agreement with historical total cumulative $CO_2$ emissions (within 1%), but RCP8.5 is also the best match out to midcentury under current and stated policies with still highly plausible levels of $CO_2$ emissions in 2100. Other works, assuming RCP8.5 is not the most-likely scenario, point out that assessing RCP8.5 might be a helpful exercise, since it flags potential risks that emerge only at the extremes (O'Neill et al., 2016).*

   *Despite the uncertainties and the limitations arising from this work (e.g. election of the forcing scenario, the use of a constant baseline mortality in all scenarios), it becomes clear from this contribution that governments and public entities must take project and clearly implement mitigation policies, which could improve air quality and therefore, the wellness of the European citizens.*

2. Although both reviewers raised a question about the model simulation and further the use of incorporating "observed" PM, the revised manuscript still lacks a clear explanation. The authors provided two previous studies (Jerez et al., 2013; Ratola and Jimenez-Guerrero, 2016), but I think these are not appropriate references for the model used in this study. [...] If the authors do not evaluate their model, the authors should provide the previous studies with the "same" model configuration (or at least the same chemical mechanism and aerosol scheme if not the same model).

   A: Following the reviewer's advice, a full model evaluation against PM2.5 observations available for Europe during the target period (taken from the AirBase database of the European Environment Agency (available at https://discomap.eea.europa.eu/App/AirQualityStatistics/index.html#) was conducted. Overall, PM2.5 data from 108 stations over Europe

was taken into account during the period 1991-2010. The results are now presented in the Supplementary Material, and the following text and figure was included in the revised version of the manuscript:

*The robustness of this simulation for representing PM2.5 is evaluated in the Supplementary Material (Tables SM2 and SM3), where the model has been compared with data from 108 stations belonging to the AirBase database of the European Environment Agency. The results are summarized in Figure 2, and the numerical results for each station can be found in Table SM3. Briefly, the low errors found (for example, average mean bias under 2 $\mu g\ m^{-3}$ and mean fractional bias < 9%) guarantee the phase accordance (timing) between the simulated and observational series, their similar amplitude and, also, the quantitative accuracy of the simulated climatologies, hence making us confident of the suitability of the modeling system for the purpose of this study.*

[Figure]

**Figure 1.** Results of the model validation for PM2.5 simulations: (a) mean bias (B, $\mu g\ m^{-3}$); (b) root mean square error (RMSE, $\mu g\ m^{-3}$); (c) mean fractional bias (MFB, %); (d) mean fractional error (MFE, %).

3. There was also no justification of using model results only, which was not constrained by satellite products or surface observations. I think there are many papers out there that used satellite products and/or surface PM2.5 observations to improve model results (Lee et al., 2015; van Donkelaar et al., 2016; Chem et al., 2020; McDuffie et al., 2021).

A: The reviewer raises a very interesting point here, related to the bias correction of the model simulations for improving the representation of PM2.5 concentrations. The assimilation or use of data to correct the bias present-climate simulations have been widely used, not only in the references mentioned by the reviewer (e.g. Lee et al., 2015; van Donkelaar et

al., 2016; Chem et al., 2020; McDuffie et al., 2021), but also by the authors of this contribution in a number of papers covering climatic periods with available observations (e.g. Jiménez-Guerrero and Ratola, 2021). However, one of the main objectives of this contribution has to do with future climate scenarios. The question arising here is: how can we correct the bias for future PM2.5, when there are no available observations to constrain the simulations? Should we use observations/satellite products to constrain present-day simulations, but not apply bias-correction techniques for the future? This could introduce an important source of uncertainty, modifying the change signal.

In order to clarify why this contribution does not use bias-correction techniques, the following discussion has been introduced in the revised version of the manuscript:

*Ground-based observations and satellite products are often used to improve modeling results for present-day simulations concerning particulate matter (e.g. Lee et al. (2015); van Donkelaar et al. (2016); Chen et al. (2020); Jiménez-Guerrero and Ratola (2021); McDuffie et al. (2021). However, these bias-correction techniques, widely used in climate impact modeling (Maraun, 2016), are limited when future scenarios are included in the simulations, since no observations can constrain future modeling results. Instead, we have decided to use the so-called "delta method" (Räisänen, 2007) to present the results and the future changes in air pollution, as recommended in Fernández et al. (2019). In the simple terms applied in this contribution, we assume that the results of the evaluation presented in the Supplementary Material point to accurate results (small biases) for present-day PM2.5 simulations, and that the difference in future (2031-2050) minus reference mean climate simulation (1991- 2010) will cancel out likely model errors. This is related to bias correction methods. In particular, delta changes are insensitive to local shift bias correction methods. It is true that more complex bias-correction techniques could have been applied (e.g. quantile mapping), but for those methods, bias corrected and delta change projections differ (Ho et al., 2012; Räisänen and Räty, 2013; Fernández et al., 2019), leading to a new source of uncertainty. Therefore, this contribution uses the delta method (assuming the cancelation of present and future biases), as also implemented in other works related to air pollution impacts on health issues (e.g. Silva et al. (2017), or the contributions of Tarín-Carrasco et al. (2019); Tarín-Carrasco et al. (2021); Guzmán et al. (2022) that rely on these very simulations; among many others).*

4. Please provide the detailed methodology of model simulation. The paper nicely presented how to calculate premature mortality and emission scenarios in detail, but does not have a model description, especially for the aerosol scheme that is critical to PM2.5 estimation. I suggest the authors include these details but are not limited to: (1) Which aerosols were simulated, by aerosol type (2) Was it sectional, bulk, or modal? How was aerosol size less than 2.5 um calculated? (3) Was nitrate aerosol included explicitly in the simulation? (4) Was secondary organic aerosol simulated? if so, which SOA scheme was used? two-product, volatility basis set, or others? what kinds of VOCs were considered for SOA precursors? (5) Was thermodynamic partitioning of aerosols calculated like Jerez et al. (2013)? If so, was it ISORROPIA or MOSAIC or other? (6) Does aerosol affect cloud and precipitation in the model?
A: The following information has been included in the revised version of the manuscript, as suggested by the reviewer:

*The parameterizations implemented in the WRF-Chem model are summarized in Table SM1 of the Supplementary Material. Further details about the methodology of the model simulations are included below. The GOCART aerosol module (Ginoux et al., 2001; Chin et al., 2002), the aerosol scheme used in this work, includes a bulk approach for black carbon (BC), organic carbon (OC), and sulfate, as well as a sectional scheme for mineral dust and sea salt using Kok (2011) brittle fragmentation theory, a simple and cheap computational approach (Palacios-Peña et al., 2020a). In this work, this scheme has been coupled with the RACM-KPP (KPP: kinetics preprocessor; Stockwell et al. (1997); Geiger et al. (2003). ISORROPIA (Nenes et al., 1998) was used for thermodynamic partitioning of aerosols.*

*In order to isolate the possible effects of climate change on pathologies only due to changes in atmospheric pollutants, constant anthropogenic emissions for all present and future simulations are assumed. Anthropogenic emissions come from the ACCMIP database (Lamarque et al., 2010) for the year 2000 by country and sector with a spatial resolution of 0.1°. This allows possible impacts to be anticipated if mitigation strategies for regulatory pollutants are not carried out and characterizes the climatic penalty on air quality levels. ACCMIP compiled a global emission dataset with annual*

*official or scientific inventories at the national or regional scale for CH$_4$, NMVOC, CO, SO$_2$, NO$_x$, NH$_3$, PM10, PM2.5, black carbon and organic carbon. Climate-dependent natural emission sources include desert dust, sea salt aerosols and biogenic volatile organic compounds (VOCs). The emissions were pre-processed according to Freitas et al. (2011).*

*As stated in Ukhov et al. (2021), the estimation of the PM2.5 is carried out by the subroutine sum_pm_gocart in module_gocart_aerosols.F. This estimation considers dust and sea salt concentration in their bins 1 (ranges 0.1-1.0 and 0.1-0.5 µm, respectively) and 2 (1.0-1.8 and 0.5-1.5 µm, respectively), black and organic carbon and sulphate. GO-CART does not include the treatment of secondary organic aerosols (SOA). The authors are aware of limitation; however, the WRF-Chem version forces to use the GOCART scheme if desert dust and sea salt aerosols are to be included (Palacios-Peña et al., 2020a). Nitrate aerosol are also not explicitly included in the simulations conducted here.*

*Last, it should be mentioned that the GOCART aerosol scheme in the WRF-Chem simulations presented here does not allow a full coupling of aerosol-cloud interactions (Palacios-Peña et al., 2020b). For instance, convective wet scavenging and cloud chemistry are not available. However, here the Morrison microphysics (Morrison et al., 2009) acts as a double moment scheme. Hence, the configuration of the model here allows a double-moment microphysics with greater flexibility when representing size distributions and hence microphysical process rates (Palacios-Peña et al., 2020a). When the double moment scheme is activated (as here), a prognostic droplet number concentration using gamma functions and mixing ratios of cloud ice, rain, snow, graupel and hail, cloud droplets, and water vapor is estimated (Morrison et al., 2009). Finally, the interaction of cloud and solar radiation with the Morrison microphysics scheme is implemented in WRF-Chem. Therefore, the droplet number will affect both the droplet mean radius and the cloud optical depth calculated by the model, affecting cloud and precipitation in the model.*

**Minor: Minor comments are mostly clarifying questions.**

1. The authors said they used climatological biomass burning emissions in response to the reviewer's comment. I fully agree with the authors' view about biomass burning emissions. It would be helpful if the authors could provide the absolute number of biomass burning emissions, especially for future studies that will compare their results to this study.
   A: As commented in the previous stage of responses to the reviewer's comments, the database for biomass burning emissions available was derived from a climatological database, and was therefore neglected in our simulations. So no biomass burning emissions was used in WRF-Chem simulations for the motives previously presented.

2. It looks like natural emission sources are different between PRE-P2010 and FUT-P2010, although anthropogenic emissions are fixed. If so, please provide the emission total of dust, sea salt, and biogenic VOCs for both present and future conditions.
   A: The reviewer is right. Natural emission sources differ between PRE-P2010 and FUT-P2010, which anthropogenic emissions fixed. The natural emissions are estimated online each timestep of the model and used internally by WRF-Chem for the calculation of PM2.5 concentrations. Unfortunately, these emissions were not stored in the project database since they were an internal input to the chemistry transport module of the model.

3. "COPD, LC, LRI, and Other NCD barely change, since these causes are not too much sensitive to PM2.5 concentration as IHD (Figure 9), [...]": Figure 9 shows similar sensitivities to PM2.5 for LRI and IHD. It needs more discussion.
   A: We agree with the reviewer's comment. The sentence has been rephrased to "On the other hand, COPD, LC, and Other NCD barely change, since these causes are not too much sensitive to PM2.5 concentration as IHD and LRI at low PM2.5 concentrations (Figure 10), as also discussed in Tarín-Carrasco et al. (2021)."